# Sign Hacking with Auxiliary Variable Exploration in the Age of Big Data

**Abstract**

In linear regression, the signs of coefficients convey the direction of covariate effects and are central to empirical interpretation. In high-dimensional settings, however, the abundance of candidate covariates introduces substantial model selection uncertainty. We study the deliberate manipulation of coefficient signs through the inclusion of a carefully chosen auxiliary variable, a practice we term SHAVE (*Sign Hacking with Auxiliary Variable Exploration*). We show that, conditional on the outcome and variables of interest, there exists a set of auxiliary-variable realizations with positive Lebesgue measure that lead to sign reversals upon inclusion. Moreover, with high probability, such variables can be found when many auxiliary candidates are available, leading simultaneously to reversed signs, inflated $t$- and $F$-statistics. Simulation studies and an empirical application corroborate these theoretical findings. We further propose detection strategies for SHAVE when augmented or independent datasets are available, as SHAVE has important implications for reproducibility, $p$-hacking, and research integrity.

**Keywords:** Sign reversal, $p$-hacking, Reproducibility, Model uncertainty, Research integrity
**Mathematics Subject Classification (2020):** 62XXX

## 1 Introduction

### 1.1 Importance of the sign of a coefficient

The sign of a regression coefficient is central to interpreting the directional effect of a variable. Sign reversals can lead to fundamentally different conclusions and policy implications. A prominent example is the Environmental Kuznets Curve (EKC), whose empirical forms—U-shaped, inverted U-shaped, N-shaped, or linear—depend critically on the estimated signs of polynomial terms in environmental covariates (Shahbaz and Sinha, 2019). Because the EKC models the relationship between economic growth and environmental degradation, such variations in coefficient signs carry substantial policy relevance for balancing economic and environmental goals. However, conflicting sign estimates introduce ambiguity and hinder the formulation of coherent sustainability policies.

### 1.2 Sign change conditions in low-dimensional linear regression

Coefficient signs in linear regression are sensitive to variable inclusion or omission. Leamer (1975) first derived a necessary condition for sign reversal based on $t$-statistics when a variable

is excluded. Subsequent studies by Visco (1978, 1988) and Oksanen (1987) provided necessary and sufficient conditions expressed through partial correlations between the omitted variable and the remaining covariates. More recently, Knaeble and Dutter (2017) established analogous conditions for sign reversals when new variables are added, formulated via inequalities involving the coefficient of determination and partial correlations. A related literature studies coefficient stability by comparing selection on observed and unobserved controls. Oster (2019) develops a robustness measure based on proportional selection, and Masten and Poirier (2026) show that reversing a coefficient sign may require substantially less unobserved selection than explaining the coefficient away. Collectively, these studies clarify the mechanisms that govern sign changes in low-dimensional linear regression settings.

## 1.3 Challenges in high-dimensional linear regression

The advent of high-dimensional data has magnified the challenges of regression analysis. Although access to a large set of covariates enhances modeling flexibility, it also increases uncertainty in selecting the final model, often destabilizing coefficient signs even in the simplest linear regression settings. First, constructing candidate models introduces new sources of uncertainty. Decisions regarding variable transformations (e.g., logarithmic or Box-Cox transformations) or basis expansions (e.g., polynomial or trigonometric terms) are typically context-dependent and rarely universally applicable. Second, conventional low-dimensional techniques perform poorly in high dimensions due to issues such as the non-invertibility of the Gram matrix. Screening procedures (Fan and Lv, 2008) can reduce dimensionality below the sample size, but often at the cost of additional screening uncertainty. Further refinement through variable selection, using methods such as Lasso (Tibshirani, 1996), Adaptive Lasso (Zou, 2006), SCAD (Fan and Li, 2001), or MCP (Zhang, 2010), also suffers from selection instability in high-dimensional applications (Nan and Yang, 2014).

The uncertainty in variable selection arises primarily from the following sources:

1. Criterion uncertainty. Different variable selection procedures employ distinct criteria, often producing divergent models and making the "best" choice unclear.

2. Cross-validation (CV) randomness. Even with a fixed method, variability in CV folds can produce unstable selection outcomes. Figure 1 illustrates this effect using the body fat dataset (Johnson, 1996), where repeated 10-fold CV leads to markedly different selected variable sets, even within the same method.

3. Slight change of data. High-dimensional selection procedures are highly sensitive to small changes in the data. For example, Nan and Yang (2014) report that removing merely 5% of observations can cause the Lasso to select models differing by more than 15 terms on average.

4. High correlations. Strong dependence among covariates is common in genomic and macroeconomic data. Dropping correlated variables risks discarding relevant predictors, whereas substituting one correlated variable for another may preserve predictive accuracy but alter interpretation. As shown in Table 1, the variable *weight* exhibits high correlations with several covariates yet is rarely selected across CV replications in Figure 1.

5. Outliers. The prevalence of outliers increases with dimensionality and no selection method is fully robust to their influence. Although many procedures incorporate outlier detection, the fundamental uncertainty in variable selection persists.

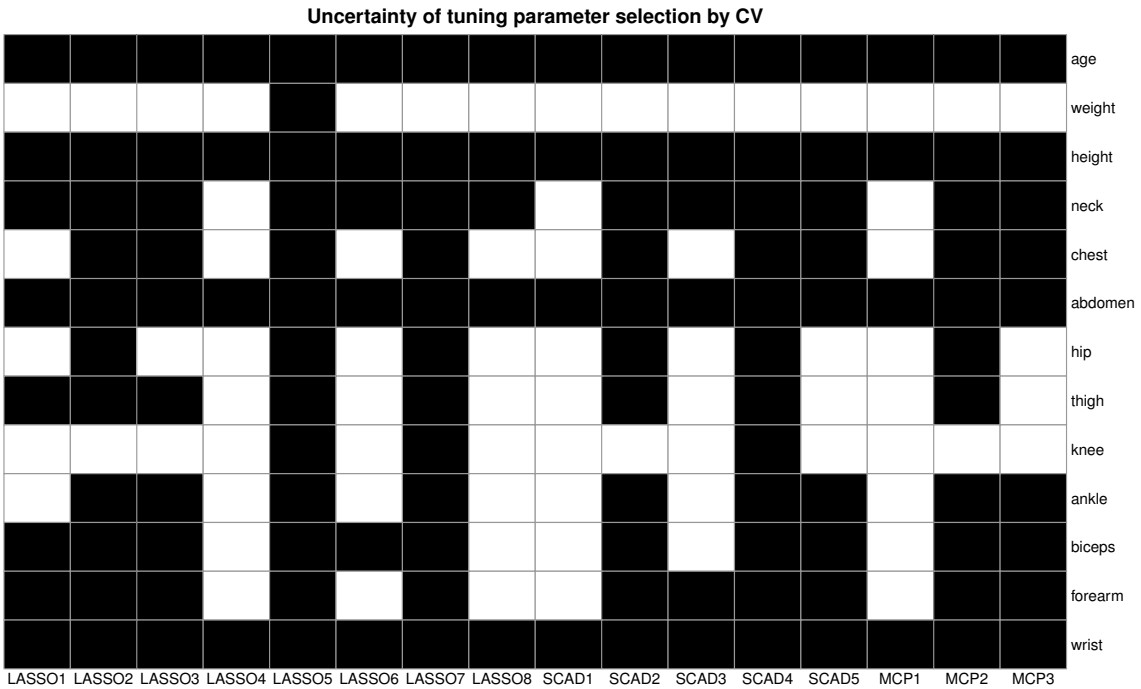

Figure 1: Model selection uncertainty in the body fat dataset. Each row denotes a covariate, and each column corresponds to a model selected by a given method, with tuning parameters determined via a single five-fold CV run. The figure displays eight Lasso models (Lasso1 to Lasso8), five SCAD models (SCAD1 to SCAD5) and three MCP models (MCP1 to MCP3). Black cells indicate covariates with nonzero coefficients in the corresponding model.

.

Existing research has sought to address the aforementioned sources of uncertainty. Methods such as VSD (Nan and Yang, 2014), SOIL importance (Ye et al., 2018), and various stability measures (Chen et al., 2007; Nogueira et al., 2017) quantify the variability of selected variables but do not reliably estimate coefficient signs. Bagging predictors (Breiman, 1996b,a) and model averaging techniques (Raftery et al., 1997; Yang and Yang, 2017; Hansen, 2007; Wan et al., 2010) reduce model selection uncertainty by aggregating across candidate models, yet their focus remains on improving prediction rather than interpretability. Another line of research develops confidence sets of models expected to contain the true specification (Hansen et al., 2011; Ferrari and Yang, 2015; Zheng et al., 2019; Lei, 2020; Li et al., 2021), explicitly acknowledging model uncertainty. However, such approaches complicate inference when coefficient signs differ across models within the confidence set. Stability selection (Meinshausen and Bühlmann, 2010; Lim and Yu, 2016) mitigates instability by retaining variables frequently selected under subsampling but still struggles under strong correlation among covariates.

## 1.4 Sign hacking

Despite these advances, uncertainty in high-dimensional model specification can still create opportunities for deliberate manipulation. Related practices, including *p*-hacking (Simmons

Table 1: Correlation matrix of the body fat data. Correlation coefficients with absolute values greater than or equal to 0.7 are shown in bold.

|  | SIRI | age | weight | height | neck | chest | abdomen | hip | thigh | knee | ankle | biceps | forearm | wrist |
|---|---|---|---|---|---|---|---|---|---|---|---|---|---|---|
| SIRI | 1 | | | | | | | | | | | | | |
| age | 0.29 | 1 | | | | | | | | | | | | |
| weight | 0.62 | -0.02 | 1 | | | | | | | | | | | |
| height | -0.03 | -0.25 | 0.51 | 1 | | | | | | | | | | |
| neck | 0.49 | 0.12 | **0.81** | 0.33 | 1 | | | | | | | | | |
| chest | **0.70** | 0.18 | **0.89** | 0.22 | **0.77** | 1 | | | | | | | | |
| abdomen | **0.82** | 0.24 | **0.87** | 0.19 | **0.73** | **0.9** | 1 | | | | | | | |
| hip | 0.63 | -0.06 | **0.93** | 0.40 | **0.71** | **0.8** | **0.86** | 1 | | | | | | |
| thigh | 0.55 | -0.22 | **0.85** | 0.35 | 0.67 | **0.7** | **0.74** | **0.88** | 1 | | | | | |
| knee | 0.49 | 0.02 | **0.84** | 0.51 | 0.65 | **0.7** | **0.71** | **0.81** | **0.78** | 1 | | | | |
| ankle | 0.25 | -0.11 | 0.58 | 0.40 | 0.43 | 0.4 | 0.41 | 0.52 | 0.50 | 0.59 | 1 | | | |
| biceps | 0.48 | -0.04 | **0.79** | 0.32 | **0.71** | **0.7** | 0.66 | **0.72** | **0.74** | 0.65 | 0.45 | 1 | | |
| forearm | 0.37 | -0.09 | 0.68 | 0.32 | 0.66 | 0.6 | 0.53 | 0.60 | 0.60 | 0.58 | 0.43 | **0.70** | 1 | |
| wrist | 0.34 | 0.22 | **0.73** | 0.40 | **0.73** | 0.6 | 0.60 | 0.63 | 0.54 | 0.66 | 0.55 | 0.61 | 0.60 | 1 |

et al., 2011; Simonsohn et al., 2014), reverse $p$-hacking (Yang and Yang, 2025), and publication bias (Lindsay, 2015; Elliott et al., 2022), exploit such uncertainty by selectively emphasizing favorable results. These practices suggest that specification uncertainty is not merely a statistical concern, but can also be strategically leveraged. This motivates the question: Can desired coefficient signs be induced by deliberately introducing observed auxiliary variables drawn from a large candidate pool? We refer to this phenomenon as "**Sign Hacking with Auxiliary Variable Exploration (SHAVE)**", an instance of unjustified manipulation of data or analysis to achieve a preferred sign pattern. When SHAVE occurs, regression results can be misleading because the hacked model may appear statistically convincing, exhibiting strong $t$-statistics and reproducible results within the hacker's constructed context. This information asymmetry between the hacker and the audience undermines scientific credibility, especially when coefficient signs are used for substantive interpretation.

This paper establishes the theoretical possibility of SHAVE in linear regression. We first show that, conditional on the outcome and variables of interest, there exists a set of auxiliary-variable realizations with positive Lebesgue measure whose inclusion induces coefficient sign reversals, offering geometric insight into when and why sign changes occur. We then demonstrate that, with high probability, one can select a variable from a large pool of auxiliary candidates that reverses the estimated signs of targeted coefficients while preserving others. Moreover, the resulting hacked coefficients can attain statistical significance and the associated model specification tests may exhibit spurious robustness. These results provide a formal mechanism underlying $p$-hacking, showing how selective specification search can generate statistically significant yet potentially spurious findings. We further discuss detection strategies based on augmented or independent datasets, as SHAVE relates to broader empirical concerns, including the reproducibility crisis, $p$-hacking, sponsorship bias and research misconduct.

The remainder of the paper is organized as follows: Section 2 develops a geometric framework that establishes the necessary and sufficient conditions for sign reversal. Section 3 introduces SHAVE and demonstrates how it can alter both the sign and statistical significance of estimated coefficients. Section 4 presents simulations and empirical evidence that support the theoretical results. Section 5 outlines potential detection strategies and Section 6 concludes.

**Notations**. We use $\|\cdot\|$ to denote the $\ell^2$ norm for vectors and the operator norm for

matrices. Vectors in Euclidean space are denoted by $\vec{\cdot}$, with inner product $\langle \vec{u}, \vec{v} \rangle = \vec{u}^\top \vec{v}$ for $\vec{u}, \vec{v} \in \mathbb{R}^n$. For two distinct vectors $\vec{u}$ and $\vec{v}$, we denote by $\text{span}\{\vec{u}, \vec{v}\}$ their linear span. The sign function $\text{sgn}(\cdot)$ is defined by $\text{sgn}\, x = 1$ if $x > 0$, $\text{sgn}\, x = -1$ if $x < 0$ and $\text{sgn}\, x = 0$ if $x = 0$. Finally, $\mathbb{1}_{\mathcal{A}}(a)$ denotes the indicator of the set $\mathcal{A}$, and we write $[a] = \{1, \ldots, a\}$ for any positive integer $a$. We denote by $P_X$ the orthogonal projection operator onto the column space of $X$, given by $P_X = X(X'X)^{-1}X'$. We further define $M_X = I - P_X$ as the projection onto the orthogonal complement of the column space of $X$. Let $I_k$ denote the $k \times k$ identity matrix.

## 2 A geometric perspective on sign changes

To understand how the signs of coefficients can be manipulated, we first characterize the geometric conditions under which sign reversals occur.

Suppose $\vec{y}, \vec{x}, \vec{v} \in \mathbb{R}^n$ are the realized sample vectors corresponding to the variables $y, x$ and $v$. The vectors $\vec{x}$ and $\vec{v}$ span a two-dimensional subspace. Regressing $y$ on $x$ and $v$ corresponds, at the sample level, to projecting $\vec{y}$ onto $\text{span}\{\vec{x}, \vec{v}\}$, yielding $P_{\vec{x}, \vec{v}}\vec{y}$. Since the signs of the estimated coefficients depend only on this projection, the analysis of sign changes can be reduced to this two-dimensional setting. We therefore focus on configurations in which $\vec{y}, \vec{x}$ and $\vec{v}$ lie in the same plane.

Without loss of generality, assume that $\vec{y}$ divides the first and second quadrants, and that $\vec{x}$ is located in the first quadrant. Under this setup, the coefficient of $x$ in the regression $y \sim x$ is positive. When $v$ is added, regressing $y$ on $x$ and $v$ corresponds geometrically to expressing $\vec{y}$ as a linear combination of $\vec{x}$ and $\vec{v}$. This representation can be visualized via a parallelogram construction: drawing lines through the endpoint of $\vec{y}$ parallel to $\vec{x}$ and $\vec{v}$, whose intersections with the lines spanned by $\vec{x}$ and $\vec{v}$ determine the signs of the corresponding regression coefficients.

By symmetry, configurations in which $\vec{v}$ lies in the third or fourth quadrant are equivalent to those in the first or second quadrant, and do not yield qualitatively different conclusions. It therefore suffices to consider cases in which $\vec{v}$ lies in the first or second quadrant. Under this restriction, only three qualitatively distinct configurations arise:

1. $\vec{v}$ in the second quadrant. Representing $\vec{y}$ in terms of $\vec{x}$ and $\vec{v}$ requires a positive coefficient on $\vec{x}$, so no sign reversal occurs.

2. $\vec{v}$ in the first quadrant between $\vec{x}$ and $\vec{y}$. In this case, $\vec{x}$ must enter with a negative coefficient to counteract the effect of $\vec{v}$, leading to a sign reversal.

3. $\vec{v}$ is in the first quadrant but forms a larger angle with $\vec{y}$ than $\vec{x}$. The contribution of $\vec{x}$ remains positive, and the sign is preserved. Note that this case is similar to the second above, except that the roles of $\vec{x}$ and $\vec{v}$ are switched.

Figure 2 illustrates these three cases, where "+" and "−" indicate sign preservation and reversal, respectively. This simple geometric argument shows that the sign reversals are driven entirely by the relative orientations of $\vec{y}, \vec{x}$ and $\vec{v}$. In Appendix B, we formalize this understanding using angles between vectors in Euclidean space, covering both planar and non-planar configurations. Expressing least squares coefficients in terms of vector angles further shows that

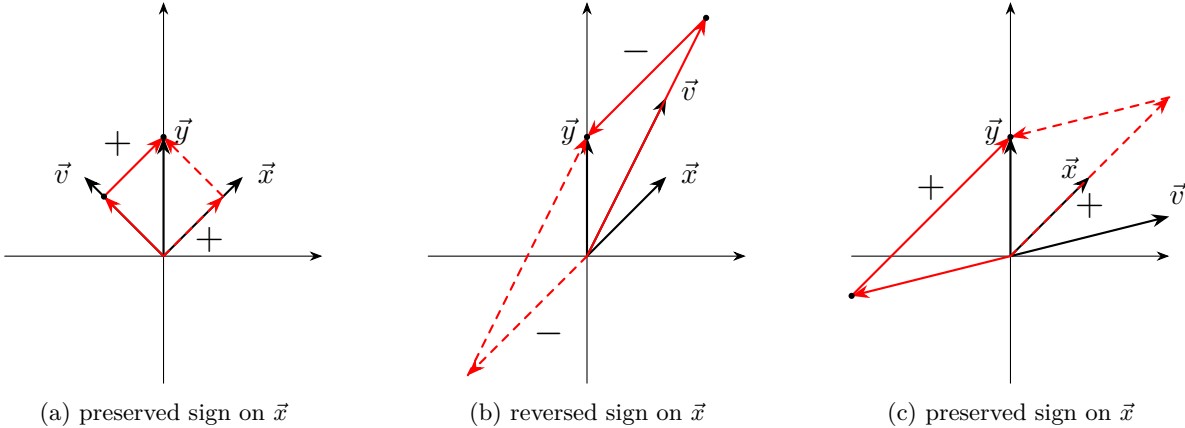

(a) preserved sign on $\vec{x}$      (b) reversed sign on $\vec{x}$      (c) preserved sign on $\vec{x}$

Figure 2: Geometric configurations for sign changes of $\vec{x}$. A "+" ("−") denotes sign preservation (reversal).

classical sign change conditions in the literature—such as those in Leamer (1975); Visco (1978, 1988) and Oksanen (1987)—are fully consistent with the geometric interpretation in Figure 2.

Building on the characterization of sign reversals for a single predictor, we now extend the analysis to the case with multiple predictors. Let $\boldsymbol{X} = (\vec{x}_1, \ldots, \vec{x}_q)$ denote a set of linearly independent vectors in $\mathbb{R}^n$. The orthogonal decomposition of $\vec{y}$ onto span($\boldsymbol{X}$) and its complement is

$$\vec{y} = \boldsymbol{X}\hat{\beta}_{\boldsymbol{X}} + \vec{e}_{\boldsymbol{X}},$$

where $\hat{\beta}_{\boldsymbol{X}} = (\boldsymbol{X}'\boldsymbol{X})^{-1}\boldsymbol{X}'\vec{y}$ and $\vec{e}_{\boldsymbol{X}} = M_{\boldsymbol{X}}\vec{y}$. We assume $\|\vec{e}_{\boldsymbol{X}}\| > 0$. After incorporating an additional vector $\vec{v}$ with $\vec{v} \notin$ span($\boldsymbol{X}$), the decomposition becomes

$$\vec{y} = \boldsymbol{X}\hat{\alpha}_{\boldsymbol{X}} + \vec{v}\hat{\alpha}_v + \vec{e}_{\boldsymbol{X}v},$$

where $\vec{e}_{\boldsymbol{X}v} = M_{\boldsymbol{X},\vec{v}}\vec{y}$ and we assume $\|\vec{e}_{\boldsymbol{X}v}\| > 0$. Let $\mathcal{I}_1$ and $\mathcal{I}_2$ form a partition of $[q]$, with either set possibly empty. The following theorem establishes the existence of a non-negligible set of perturbation vectors capable of inducing selective sign reversals within $\boldsymbol{X}$.

**Theorem 2.1.** *Given* $(\vec{y}, \boldsymbol{X})$, *there exists a measurable set* $\mathcal{V}^* \subset \mathbb{R}^n$ *with positive Lebesgue measure such that for every* $\vec{v} \in \mathcal{V}^*$,

$$\operatorname{sgn}\hat{\beta}_{\boldsymbol{X},j} = -\operatorname{sgn}\hat{\alpha}_{\boldsymbol{X},j}, \quad j \in \mathcal{I}_1 \ \text{and} \ \operatorname{sgn}\hat{\beta}_{\boldsymbol{X},j} = \operatorname{sgn}\hat{\alpha}_{\boldsymbol{X},j} \quad j \in \mathcal{I}_2. \tag{2.1}$$

*A constructive realization of such a set can be obtained by considering vectors of the form*

$$\vec{v} = \boldsymbol{X}A + b\vec{y} + \varepsilon\vec{u}.$$

*Choose* $A \in \mathbb{R}^q$ *with* $A_j \neq 0$ *for all* $j \in [q]$ *and define*

$$\kappa_{\min}(A) = \min_{j\in[q]} \frac{|A_j| \cdot \|M_{\boldsymbol{X}_{-j}}\boldsymbol{X}_j\|}{\|M_{\boldsymbol{X}_{-j}}\vec{y}\|}, \quad \kappa_{\max}(A) = \max_{j\in[q]} \frac{|A_j| \cdot \|M_{\boldsymbol{X}_{-j}}\boldsymbol{X}_j\|}{\|M_{\boldsymbol{X}_{-j}}\vec{y}\|}. \tag{2.2}$$

*Fix any $b > 0$ satisfying $b \geq \kappa_{\max}(A)$. The choices of $A$ and $b$ are required to satisfy*

$$bA_j\hat{\beta}_{\boldsymbol{X},j} > 0 \text{ for } j \in \mathcal{I}_1, \quad bA_j\hat{\beta}_{\boldsymbol{X},j} < 0 \text{ for } j \in \mathcal{I}_2. \tag{2.3}$$

*For some fixed $\eta \in (0, 1/4)$, let $\varepsilon_0 = \eta\|M_{\boldsymbol{X}}\vec{y}\|\kappa_{\min}(A)$. Define*

$$\mathcal{V} = \{\vec{v} = \boldsymbol{X}A + b\vec{y} + \varepsilon\vec{u} : \vec{u} \in \mathbb{S}^{n-1}, 0 < \varepsilon \leq \varepsilon_0\}. \tag{2.4}$$

*Here $\boldsymbol{X}_{-j}$ denotes the design matrix obtained from $\boldsymbol{X}$ by removing its $j$-th column. In particular, one may take $\mathcal{V}^* = \mathcal{V}$.*

**Remark 2.1.** Once a sign convention for $b$ has been fixed, condition (2.3) imposes only sign restrictions on $A$, and is therefore readily satisfied. Given $(\vec{y}, \boldsymbol{X})$, the signs of $\hat{\beta}_{\boldsymbol{X}}$ are fixed, so admissible choices of $A$ always exist. The magnitudes of $A$ and $b$ determine the admissible range of $\varepsilon$ and hence the size of the constructed set $\mathcal{V}$.

**Remark 2.2.** Theorem 2.1 accommodates all possible patterns of selective sign change, ranging from a single reversal to simultaneous reversals across all coefficients, while allowing the remaining coefficients to retain their original signs. For example, taking $\mathcal{I}_1$ as a singleton and $\mathcal{I}_2 = \mathcal{I}_1^c$ yields a single sign reversal, whereas setting $\mathcal{I}_1 = [q]$ and $\mathcal{I}_2 = \emptyset$ produces simultaneous reversals of all coefficients. As the number of imposed sign-reversal constraints increases, the admissible set $\mathcal{V}$ becomes more restricted.

**Remark 2.3.** Theorem 2.1 generalizes classical sign reversal phenomena such as Simpson's paradox (Simpson, 1951) and negative suppression (Nickerson and Brown, 2019). In contrast to the bivariate setting, it accommodates configurations in which multiple coefficients reverse simultaneously while others remain unchanged. The existence of a measurable set $\mathcal{V}^*$ with positive Lebesgue measure shows that such selective reversals occur over a non-negligible region of the sample space, rather than occurring only in degenerate or asymptotic cases.

The proposed sign change condition holds for any fixed sample size $n$, ensuring its relevance for empirical applications. Formulated in Euclidean space under the Lebesgue measure, it provides a geometric characterization of the set of vectors $\vec{v}$ that induce sign reversals. To the best of our knowledge, it is the first condition that enables both identifying and quantifying this set within a unified geometric framework. This perspective yields a transparent and interpretable approach for understanding the mechanisms underlying SHAVE, particularly in high-dimensional settings where existing analytic conditions become intractable under selection uncertainty. The resulting characterization will serve as the foundation for analyzing deliberate sign manipulation in Section 3.

## 3  The SHAVE

Recall that SHAVE refers to the introduction of auxiliary variables to achieve a desired sign pattern of estimated coefficients. This phenomenon has been documented in a range of empirical settings, as illustrated by the following examples.

**Example 1** (Covariate manipulation)**.** Diederik Stapel, a former professor of social psychology at Tilburg University, was found guilty of fabricating and manipulating data over several years. One technique involved covariate manipulation, where variables were selectively included in or excluded from regression models to produce desired outcomes[1]. Similarly, Brian Wansink, formerly at Cornell University, engaged in selective reporting and covariate adjustments to obtain favorable outcomes[2].

Although SHAVE is framed from the perspective of deliberate manipulation, similar patterns also arise unintentionally under exploratory model specification.

**Example 2** (Determinants of Economic Growth)**.** The long-standing debate on the determinants of economic growth illustrates how model specifications can alter coefficient signs even in the absence of intentional manipulation. Levine and Renelt (1992) showed that regression outcomes in cross-country growth studies are highly sensitive to variable inclusion, with coefficient signs varying across specifications. Typical regressors include variables such as *imports, openness measures, credit growth, and political instability indicators*. Sala-i Martin et al. (2004) further evaluated 67 commonly used covariates using Bayesian Averaging of Classical Estimates. Among the top 18 variables identified, half exhibited sign reversals in more than 2% of posterior draws. Their analysis, based on nearly 89 million regressions, underscores the instability of estimated relationships in this literature.

These examples motivate a theoretical investigation into the mechanisms underlying SHAVE and its implications for statistical inference in finite samples.

Now suppose a hacker collects an original sample $(y_i, x_i)$ of size $n$ and obtains the baseline regression model:

$$Y = X\hat{\beta}_X + \hat{e}_X, \tag{3.1}$$

where $Y = (y_1, \ldots, y_n)'$, $X = (x_1', \ldots, x_n')'$ and each $x_i \in \mathbb{R}^q$. Here, $\hat{\beta}_X$ is the OLS estimate and $\hat{e}_X$ is the associated residual vector. The analysis is conducted in a finite-sample setting with fixed $q$, assuming $X$ has full column rank and $\|\hat{e}_X\| > 0$ to exclude degenerate cases.

Some estimated coefficients in $\hat{\beta}_X$ may not exhibit the signs desired by the hacker. To alter these signs, the hacker introduces an additional covariate drawn from a *auxiliary* pool of variables $X_A \in \mathbb{R}^{n \times p}$, complementing the original covariate set $X$. Let $V \in X_A$ denote a candidate covariate. The hacker then fits the augmented model:

$$Y = X\hat{\alpha}_X + V\hat{\alpha}_V + \hat{e}_{XV}, \tag{3.2}$$

where $\hat{\alpha}_X$ and $\hat{\alpha}_V$ are the corresponding OLS estimates. Denote the target sign configuration by $\text{sgn}\,\hat{\alpha}_X = S^\star$, where $S^\star \in \{-1, 1\}^q$ represents the intended sign pattern and differs from $\text{sgn}\,\hat{\beta}_X$ in at least one component. Achieving this configuration constitutes a successful instance of SHAVE.

In low-dimensional settings where the auxiliary pool $X_A$ is finite and limited in size, the effect of each candidate covariate can, in principle, be assessed using classical sign change conditions.

---

[1] See "Flawed science: The fraudulent research practices of social psychologist Diederik Stapel"

[2] See "Here's How Cornell Scientist Brian Wansink Turned Shoddy Data Into Viral Studies About How We Eat"

Since the set of admissible perturbations is small, such manipulations are relatively transparent and easier to detect. In contrast, in high-dimensional environments, the auxiliary pool may contain a vast number of candidate covariates, many of which are correlated with both $X$ and $Y$. This substantially enlarges the space of feasible perturbations. With a sufficiently rich pool, the hacker can often rationalize the inclusion of a selected $V$ as the outcome of a legitimate data-driven procedure, such as variable selection. Consequently, even when a sign reversal is observed, it becomes difficult to determine whether it reflects genuine statistical evidence or deliberate specification search. This difficulty arises from an intrinsic identification problem driven by the interaction between model uncertainty and high-dimensional search.

The hacker can construct a large auxiliary pool $X_A$ through several generic mechanisms that are widely available in practice:

1. Transformations of existing variables. This includes polynomial expansions and interaction terms of the original covariates $X$, which are commonly used to capture nonlinearities and interactions. Such constructions are ubiquitous across applications, including marketing analytics, social networks (Hao and Zhang, 2014), and genomics (Cordell, 2009).

2. Incorporation of additional structured covariates. New variables can be introduced from extended or alternative data sources, such as different measures of inputs in productivity analysis (Syverson, 2011), additional determinants in growth regressions (Sala-i Martin et al., 2004; Ciccone and Jarociński, 2010), or supplementary probes in genomic studies (Dufva, 2005).

3. Feature extraction from unstructured data. Covariates can also be constructed from text, images, or network representations. These approaches have become increasingly common in modern empirical work, as illustrated by applications involving social media content (Erevelles et al., 2016), political speeches (Gentzkow et al., 2019), product attributes in marketing (Wang et al., 2022), or graph- or image-based features (Yan et al., 2015; Shuman et al., 2013).

These strategies substantially expand the set of admissible covariates, often generating a high-dimensional and highly redundant pool $X_A$. This enlarged pool provides a rich collection of directions along which the baseline regression can be perturbed, thereby increasing the likelihood of achieving a desired sign pattern. Given the ease of constructing such large auxiliary pools, this raises the following question: Can the hacker identify a covariate $V$ whose inclusion induces a sign change in the target coefficients, thereby achieving the desired sign pattern through SHAVE? To address this question, it is necessary to formalize the richness of the auxiliary pool $X_A$. The following conditional independence assumption characterizes a setting in which sufficiently diverse directions are available for such perturbations, ensuring that the search over $V$ is not restricted to a narrow subset of the sample space.

**Assumption 1** (Conditional Independence)**.** There exists a subset of indices $\mathcal{J} \subseteq [p]$ with $|\mathcal{J}| = p_1$ such that, conditional on $(Y, X)$, the variables $\{X_{A,j}, j \in \mathcal{J}\}$ are mutually independent. Moreover, for each $j \in \mathcal{J}$, the normalized variable $\frac{X_{A,j}}{\|X_{A,j}\|}$ has a density uniformly bounded away from zero on the unit sphere.

**Remark 3.1.** The conditional independence assumption can be relaxed. If the variables in $X_A$ exhibit mild dependence, such as temporal or index-wise correlation satisfying suitable mixing conditions, the subsequent results continue to hold with additional technical adjustments.

Assumption 1 ensures that the realizations of variables in $X_A$ are sufficiently dispersed over the unit sphere, in the sense that directions generated by a subset of auxiliary variables cover the space without degeneracy (see Lemma A.1 in the Appendix A.2). A key feature of this assumption is that it imposes minimal restrictions on the overall dependence structure within $X_A$. Only a subset of size $p_1$ is required to exhibit conditional independence, while the remaining variables may be arbitrarily dependent. Consequently, the condition is substantially weaker than classical high-dimensional regularity assumptions such as the irrepresentable condition or the restricted isometry property.

Under this condition, the auxiliary pool provides a sufficiently rich collection of directions, so that the hacker's search over $V \in X_A$ becomes increasingly effective as the pool size grows. Under Assumption 1, as the size of the auxiliary pool increases, the hacker can identify at least one variable $V \in X_A$ that induces the desired sign pattern with probability approaching one. Let $\mathcal{I}_1, \mathcal{I}_2 \subseteq [q]$ denote two fixed complementary index sets.

**Theorem 3.1.** *Suppose Assumption 1 holds. Then, given $(Y, X)$, as $p_1 \to \infty$, with probability tending to one, there exists at least one $V \in X_A$ such that*

$$\operatorname{sgn} \hat{\beta}_{\boldsymbol{X},j} = -\operatorname{sgn} \hat{\alpha}_{\boldsymbol{X},j}, \quad j \in \mathcal{I}_1 \ and \ \operatorname{sgn} \hat{\beta}_{\boldsymbol{X},j} = \operatorname{sgn} \hat{\alpha}_{\boldsymbol{X},j} \quad j \in \mathcal{I}_2. \tag{3.3}$$

**Remark 3.2.** The inclusion of the auxiliary variable $V$ also determines the sign of the auxiliary coefficient. Under the normalization $b > 0$, the coefficient on $V$ satisfies $\hat{\alpha}_V > 0$, or equivalently $\operatorname{sgn} \hat{\alpha}_V = \operatorname{sgn} \frac{1}{b}$. This sign convention is immaterial for the main result. Replacing $V$ by $-V$ leaves the column space spanned by $(X, V)$ unchanged and therefore leaves the coefficients on $X$ unchanged, while it reverses only the coefficient on the auxiliary variable. Thus the normalization $b > 0$ is made without loss of substantive content for Theorem 3.1.

**Remark 3.3** (Invariance to Model Perturbation)**.** Given $(Y, X)$, the constructed set $\mathcal{V}$ inducing sign changes is deterministic. Although membership in $\mathcal{V}$ provides only a sufficient condition for sign change, it plays the same role for both directions of model perturbation. Starting from a model without $V$, adding a variable whose realization lies in $\mathcal{V}$ induces a sign change; likewise, starting from a model that includes $V$, omitting such a variable also induces a sign change. In this sense, the conditions for sign change under inclusion and omission are characterized by the same geometric object determined by $(Y, X)$. This yields a unified geometric perspective, under which existing conditions in the literature can be viewed as describing different subsets or special cases of this construction.

**Remark 3.4.** Although selection consistency guarantees asymptotic identification of relevant variables, it offers limited protection against sign manipulation in finite samples. In practice, sample sizes are often insufficient for such asymptotic guarantees to be effective. Moreover, the information asymmetry between the hacker and the audience allows the hacker to conceal the set of explored variables, making such manipulation intrinsically difficult to detect.

Beyond altering the signs of coefficients, the hacker can further identify a variable $V$ such that the resulting coefficients $\hat{\alpha}_X$ are statistically significant, thereby making SHAVE more deceptive with high probability. This yields a positive-measure refinement of the sign change set $\mathcal{V}$.

**Theorem 3.2.** *Let $\delta \in (0,1)$ be a given significance level. For any constant $c > 0$, there exists a subset $\mathcal{V}_{\delta,c} \subseteq \mathcal{V}$ with positive Lebesgue measure such that, for any $V \in \mathcal{V}_{\delta,c}$, the sign change relation* (3.3) *holds and*

$$\min_{j \in [q]} |T_j| \geq c \cdot t_{n-q-1,\delta/2},$$

*where $T_j$ denotes the t-statistic associated with $\hat{\alpha}_{X,j}$. Furthermore, under Assumption 1, with probability going to one as $p_1 \to \infty$, there exists at least one $V \in X_A$ satisfying these properties.*

**Remark 3.5.** While it is possible to find a variable $V$ that produces arbitrarily large $t$-statistics, such extreme values are typically not required. In practice, it suffices to identify a variable that pushes the $t$-statistic just beyond the critical threshold. This type of marginal manipulation reflects the phenomenon of $p$-hacking, as discussed in Appendix C. Results obtained in this way can appear more convincing precisely because the induced significance is subtle rather than extreme.

While the sign change region ensures selective reversals and statistical significance in $\hat{\alpha}_X$, a further refinement yields a positive-measure subset on which the auxiliary coefficient $\hat{\alpha}_V$ is also statistically significant. Moreover, under Assumption 1, such variables can be found within $X_A$ with probability tending to one. In particular, this refinement is taken within $\mathcal{V}_{\delta,c}$, so that both sets of $t$-statistic conclusions hold simultaneously on the resulting subset.

**Corollary 3.2.1.** *Under the same conditions as in Theorem 3.2, for any constant $c > 0$, there exists a subset $\mathcal{V}^1_{\delta,c} \subseteq \mathcal{V}_{\delta,c}$ with positive Lebesgue measure such that, for any $V \in \mathcal{V}^1_{\delta,c}$, the inclusion of $V$ satisfies $\operatorname{sgn} \hat{\alpha}_V = \operatorname{sgn} \frac{1}{b}$ and*

$$|T_V| \geq c \cdot t_{n-q-1,\delta/2},$$

*where $T_V$ denotes the t-statistic associated with $\hat{\alpha}_V$. Furthermore, under Assumption 1, as $p_1 \to \infty$, with probability tending to one, there exists at least one $V \in X_A$ satisfying these properties.*

Model specification tests, such as the $F$-test and the Wald test, are commonly used to assess whether groups of regression coefficients are jointly equal to zero, thereby supporting or refuting a given model specification. The hacker can exploit these tests by selecting a variable $V$ such that any given subvector of $\alpha_X$ appears statistically significant, leading to rejection of the corresponding null hypothesis with high probability. This further enhances the apparent credibility of the manipulated model. A further refinement can be constructed within $\mathcal{V}^1_{\delta,c}$, thereby ensuring that the sign change relation, the lower bounds on the $t$-statistics for both $\hat{\alpha}_X$ and $\hat{\alpha}_V$, and the desired $F$-test rejection all hold simultaneously.

**Corollary 3.2.2.** *Let $I \subseteq \mathcal{I}_1 \cup \mathcal{I}_2$ be any nonempty index set with $|I| = s$ and consider the augmented model* (3.2). *For any significance level $\delta \in (0,1)$ and any constant $c > 0$, there exists*

*a measurable subset $\mathcal{V}_{\delta,I,c} \subseteq \mathcal{V}^1_{\delta,c}$ with positive Lebesgue measure such that, for every $V \in \mathcal{V}_{\delta,I,c}$, the sign change relations (3.3) hold and the F-statistic for testing $H_0 : \alpha_{X,I} = 0$ satisfies*

$$F_I \geq c \cdot F_{\delta;s,n-q-1}.$$

*Furthermore, under Assumption 1, as $p_1 \to \infty$, with probability approaching one, there exists at least one $V \in X_A$ satisfying these properties.*

**Remark 3.6.** For any $V \in \mathcal{V}_{\delta,I,c}$, an $F$-test comparing models with and without $V$ rejects the null hypothesis that the coefficient of $V$ is zero, thereby favoring the inclusion of $V$, due to the equivalence between the $t$- and $F$-tests for a single linear restriction.

Based on the preceding results, the hacker implements SHAVE in a stylized manner. A baseline regression is first estimated using covariates selected according to prior beliefs, existing studies or standard variable selection procedures. When the resulting coefficients are deemed unsatisfactory, the model is subsequently manipulated by searching for an auxiliary variable $V$ that delivers the desired outcome, as guaranteed by Theorem 2.1. Incorporating such a $V$ ensures that the coefficients of interest attain the targeted signs and sufficiently large $t$-statistics, as established by Theorems 3.1 and 3.2. To further enhance apparent credibility, the hacker conducts $F$-tests on subsets of coefficients to validate the model specification, which, under SHAVE, can be mechanically satisfied by Corollary 3.2.2.

This implementation of SHAVE is closely related to several well-documented empirical phenomena. By inducing sample dependence between the auxiliary variable $V$, the outcome $Y$ and the regressors $X$, SHAVE can reverse the estimated sign of coefficients of $X$ even when $V$ has no causal effect on $Y$, exemplifying **spurious correlation** (Aldrich, 1995; Fan et al., 2014). When such results are selectively reported, the resulting inflation of $t$-statistics or related test statistics reflects $p$-**hacking** (Simmons et al., 2011) and contributes to **publication bias** (Sterling, 1959). Moreover, post hoc analyses conducted after SHAVE that systematically favor particular conclusions may introduce **sponsorship bias** (Lesser et al., 2007), and in extreme cases, can raise concerns related to **research integrity**. Prominent examples of misleading or fabricated findings, such as the stem-cell and STAP cell scandals, illustrate how such practices can distort the scientific record. A detailed discussion of these connections is provided in the Appendix C.

Despite its seemingly sound statistical presentation, the model produced by the hacker is fundamentally misleading. The auxiliary variable $V$ is selected precisely because its realization happens to satisfy the sign change condition, rather than due to any substantive relationship with the outcome. As a result, the apparent statistical evidence arises from targeted specification search rather than underlying structure. This issue may arise even in settings typically viewed as robust, such as randomized controlled trials (RCTs). In practice, full randomization is often infeasible, particularly in applications such as personalized medicine, where treatments are tailored to individual characteristics. In such settings, patient attributes, such as genetic markers or pre-existing conditions, may be correlated with treatment assignment. Conditioning on covariates selected from these attributes can therefore introduce distortions in treatment effect estimation. Senn (2018) highlights related concerns, which can be viewed as practical manifestations of SHAVE.

# 4 Numerical examples

In this section, numerical experiments are used to complement the theoretical analysis. We first examine whether SHAVE can be effectively implemented by the hacker. We then demonstrate that the constructed set $\mathcal{V}$ is sufficient to induce sign changes, as discussed in Section 2. Finally, we illustrate how SHAVE may manifest in practice from the hacker's perspective using a rat eye gene expression dataset.

## 4.1 Feasibility of SHAVE

To motivate the simulation from a concrete hacker's perspective, we consider a stylized nutrition setting inspired by the sponsorship-bias literature.

Suppose a hacker backed by a food-industry sponsor with commercial interests in a particular dietary product purports to study the effects of the sponsored product ($x_1$) and fat intake ($x_2$) on an adverse cardiovascular outcome ($y$), such as serum LDL cholesterol or a coronary heart disease risk index. The hacker seeks a significantly negative coefficient on the sponsored product, together with a significantly positive coefficient on fat, thereby supporting the preferred narrative that fat, rather than the sponsored product, is the primary dietary culprit. He begins with the baseline regression of $y$ on $x_1$ and $x_2$, based on an independent sample $(y_i, x_{1i}, x_{2i})_{i=1}^n$ generated from

$$y_i = x_{1i}\beta_1 + x_{2i}\beta_2 + e_i, \quad e_i \sim N(0, \sigma^2), \tag{4.1}$$

where $n \in \{20, 30, 100\}$, $(\beta_1, \beta_2) \in \{(0.2, 0.2), (0.5, 0.5)\}$ and $\sigma^2 = 0.5$. Thus, by construction, both exposures have positive coefficients, corresponding to harmful effects on the adverse health outcome. The regressors $(x_{1i}, x_{2i})$ follow a bivariate normal distribution with mean zero and covariance matrix $\Sigma$ with entries $\Sigma_{ij} = \rho^{|i-j|}$, where $\rho \in \{0.1, 0.5, 0.9\}$.

Let $Y = (y_1, \ldots, y_n)'$, $X_1 = (x_{11}, \ldots, x_{1n})'$, $X_2 = (x_{21}, \ldots, x_{2n})'$ and $\hat{e}_X$ be the residual vector from the baseline regression

$$Y = X_1\hat{\beta}_1 + X_2\hat{\beta}_2 + \hat{e}_X. \tag{4.2}$$

When the baseline estimates have the correct signs, both $\hat{\beta}_1$ and $\hat{\beta}_2$ are positive, which does not support the sponsor's preferred narrative. The hacker therefore searches over a large auxiliary pool $X_A$ for an additional covariate $V$ whose inclusion reverses the estimated effect of the sponsored product while preserving the positive effect of fat.

In substantive terms, $X_A$ represents a rich collection of plausible auxiliary variables available to the hacker, including alternative dietary measures, behavioral indicators, health-related covariates, transformations of existing variables (possibly including linear combinations of $X_1$ and $X_2$) and other constructed features. In practice, such features may also be correlated with the original exposures. In the simulation, however, we abstract from these domain-specific details and idealize the high-dimensional search environment by generate $10^8$ candidate variables independently from a standard normal distribution. This construction satisfies Assumption 1. The hacker then searches over $X_A$ to identify a variable $V$ such that, in the augmented model

$$Y = X_1\hat{\alpha}_1 + X_2\hat{\alpha}_2 + V\hat{\alpha}_V + \hat{e}_{XV}, \tag{4.3}$$

the coefficient on the sponsored product is negative and statistically significant, while that on fat is positive and statistically significant, both at the 10% level.

Within this framework, the following quantities are reported. First, although $(\hat{\beta}_1, \hat{\beta}_2)$ are expected to recover the sign pattern of $(\beta_1, \beta_2)$, sampling variability, especially in small samples, may lead to sign discrepancies. We therefore report the probability with which the baseline estimates $(\hat{\beta}_1, \hat{\beta}_2)$ match the true sign pattern, denoted as "Valid", to quantify sampling variability. Conditional on this event, we define SHAVE as the occurrence of the target sign configuration in the augmented model, given by $\text{sgn}\,\hat{\alpha}_1 = -\,\text{sgn}\,\hat{\beta}_1$ and $\text{sgn}\,\hat{\alpha}_2 = \text{sgn}\,\hat{\beta}_2$, and report its probability, denoted by "SHAVE". We also report the probability that there exists at least one auxiliary variable $V \in X_A$ that induces SHAVE ("Found"). In addition, we report the conditional probabilities with which $\hat{\alpha}_1$ and $\hat{\alpha}_2$ are statistically significant at the 10% level when SHAVE occurs, denoted as Sig.$(\alpha_1)$ and Sig.$(\alpha_2)$, to assess the extent to which the manipulated results appear statistically convincing. All results are computed based on the first $10^6, 10^7$ and $10^8$ variables in $X_A$, averaged over 100 Monte Carlo replications.

Table 2: SHAVE frequency under independent auxiliary variables.

| $n$ | $(\beta_1, \beta_2)$ | $\rho$ | Valid | FOUND (%) | | | SHAVE (%) | | | Sig.$(\alpha_1)$ (%) | | | Sig.$(\alpha_2)$ (%) | | |
|---|---|---|---|---|---|---|---|---|---|---|---|---|---|---|---|
| | | | | $10^6$ | $10^7$ | $10^8$ | $10^6$ | $10^7$ | $10^8$ | $10^6$ | $10^7$ | $10^8$ | $10^6$ | $10^7$ | $10^8$ |
| 20 | (0.2,0.2) | 0.1 | 73% | 94.5 | 98.6 | 100.0 | 1.989 | 1.991 | 1.992 | 0.034 | 0.052 | 0.047 | 43.196 | 44.882 | 45.922 |
| | | 0.5 | 71% | 98.6 | 98.6 | 98.6 | 2.820 | 2.822 | 2.821 | 0.045 | 0.056 | 0.057 | 53.926 | 54.092 | 54.074 |
| | | 0.9 | 40% | 100.0 | 100.0 | 100.0 | 6.370 | 6.374 | 6.375 | 0.063 | 0.046 | 0.053 | 26.835 | 26.622 | 26.797 |
| | (0.5,0.5) | 0.1 | 100% | 56.0 | 80.0 | 89.0 | 0.236 | 0.236 | 0.236 | 0.022 | 0.525 | 0.076 | 49.152 | 71.086 | 78.163 |
| | | 0.5 | 97% | 76.3 | 91.8 | 95.9 | 0.072 | 0.072 | 0.072 | 0.040 | 0.033 | 0.049 | 73.039 | 87.609 | 92.824 |
| | | 0.9 | 79% | 100.0 | 100.0 | 100.0 | 2.752 | 2.750 | 2.750 | 0.060 | 0.052 | 0.064 | 86.883 | 86.727 | 86.809 |
| 30 | (0.2,0.2) | 0.1 | 86% | 81.4 | 91.9 | 97.7 | 0.754 | 0.753 | 0.753 | 0.004 | 0.006 | 0.005 | 48.993 | 56.119 | 58.756 |
| | | 0.5 | 82% | 85.4 | 93.9 | 97.6 | 1.660 | 1.659 | 1.659 | 0.003 | 0.005 | 0.005 | 63.076 | 69.472 | 72.517 |
| | | 0.9 | 49% | 100.0 | 100.0 | 100.0 | 3.361 | 3.361 | 3.362 | 0.010 | 0.010 | 0.010 | 36.681 | 36.484 | 36.497 |
| | (0.5,0.5) | 0.1 | 100% | 15.0 | 22.0 | 36.0 | 0.000 | 0.000 | 0.000 | 0.000 | 0.001 | 0.002 | 13.811 | 21.090 | 34.965 |
| | | 0.5 | 100% | 24.0 | 38.0 | 50.0 | 0.109 | 0.109 | 0.109 | 0.000 | 0.005 | 0.001 | 23.750 | 37.973 | 49.724 |
| | | 0.9 | 91% | 73.6 | 94.5 | 95.6 | 0.894 | 0.894 | 0.893 | 0.002 | 0.095 | 0.027 | 73.625 | 94.414 | 95.447 |
| 100 | (0.2,0.2) | 0.1 | 98% | 8.2 | 13.3 | 17.3 | 0.000 | 0.000 | 0.000 | 0.000 | 0.000 | 0.000 | 6.461 | 11.353 | 15.396 |
| | | 0.5 | 98% | 12.2 | 20.4 | 28.6 | 0.163 | 0.163 | 0.163 | 0.000 | 0.000 | 0.000 | 12.245 | 20.408 | 28.569 |
| | | 0.9 | 83% | 55.4 | 67.5 | 75.9 | 0.174 | 0.174 | 0.175 | 0.000 | 0.000 | 0.000 | 52.102 | 63.697 | 70.722 |
| | (0.5,0.5) | 0.1 | 100% | 0.0 | 0.0 | 0.0 | 0.000 | 0.000 | 0.000 | 0.000 | 0.000 | 0.000 | 0.000 | 0.000 | 0.000 |
| | | 0.5 | 100% | 0.0 | 0.0 | 0.0 | 0.000 | 0.000 | 0.000 | 0.000 | 0.000 | 0.000 | 0.000 | 0.000 | 0.000 |
| | | 0.9 | 100% | 3.0 | 8.0 | 11.0 | 0.002 | 0.003 | 0.003 | 0.000 | 0.000 | 0.000 | 3.000 | 8.000 | 11.000 |

Note: "Valid" denotes the event that both baseline coefficients have the correct signs, i.e., $\text{sgn}\,\hat{\beta}_j = \text{sgn}\,\beta_j$ for $j = 1, 2$. "FOUND" denotes the probability that there exists at least one $V \in X_A$ that induces SHAVE. "SHAVE" reports the probability that $\text{sgn}\,\hat{\alpha}_1 = -\,\text{sgn}\,\hat{\beta}_1$ and $\text{sgn}\,\hat{\alpha}_2 = \text{sgn}\,\hat{\beta}_2$, conditional on "Valid". Sig.$(\alpha_1)$ and Sig.$(\alpha_2)$ denote the proportion of statistically significant estimates (10% level) for $\hat{\alpha}_1$ and $\hat{\alpha}_2$, respectively, conditional on "SHAVE". Entries are percentages based on $10^6$ – $10^8$ generated realizations of auxiliary variables over 100 Monte Carlo replications.

Table 2 reports the results. The probability of identifying at least one auxiliary variable that induces SHAVE increases with the number of candidate variables and approaches one in many configurations, as shown in the column "FOUND". At the same time, SHAVE occurs with positive frequency across nearly all configurations, and these frequencies remain essentially unchanged across different sizes of auxiliary variables, as indicated by column "SHAVE". The effect of sample size is especially clear: as $n$ increases, the frequency of SHAVE declines substantially, indicating that larger samples make the hacking task much more difficult. A stronger signal has a similar stabilizing effect. Conditional on SHAVE events, both $\hat{\alpha}_1$ and $\hat{\alpha}_2$ exhibit nontrivial probabilities of statistical significance at the 10% level. These probabilities generally increase as more auxiliary variables are searched, although the significance frequency of the sign-reversing

coefficient $\hat{\alpha}_1$ is more variable in small samples. Note that this illustration sets a tougher goal of simultaneously enforcing the sign of $X_1$ and $X_2$. Due to computational constraints, we restrict the search to at most $10^8$ auxiliary variables. With more and more variables searched, the hacking success probability will eventually be 1.

The role of correlation is more subtle. The increase in SHAVE under stronger correlation does not align with the construction of the set $\mathcal{V}$ in Eq. (2.4), under which stronger dependence shrinks the admissible perturbation region. This discrepancy arises because $\mathcal{V}$ characterizes only a sufficient subset of perturbations that induce sign changes. A full characterization of the necessary set for sign reversal, particularly how dependence among regressors shapes the geometry of the joint sign change region, requires further analysis and is left for future work.

Note that the original estimates do not always recover the true sign pattern, as the frequency of "Valid" falls below one in some configurations. This implies that the baseline regression may occasionally produce coefficient estimates that align with the hacker's desired signs due to sampling variability, even when they contradict the true parameter values, potentially creating a misleading impression without any manipulation (though it is unlikely the coefficients are significant). In this case, based on our theoretical results, the hacker can still search for an auxiliary variable $V$ that preserves the sign pattern while boosting their significance. When the baseline results contradict the intended narrative, the hacker resorts to SHAVE to achieve suitable signs. The findings above provide numerical support for the mechanism underlying SHAVE. In particular, the non-negligible frequency of SHAVE events confirms that a hacker can, in practice, identify auxiliary variables that overturn unfavorable results. The observed patterns further indicate that such manipulation becomes easier in environments with strong dependence and limited information. Moreover, the prevalence of statistically significant outcomes conditional on SHAVE implies that the resulting models can appear highly credible, reinforcing the practical relevance of the proposed mechanism.

## 4.2 Sufficiency of the constructed set $\mathcal{V}$

The preceding simulation documents the practical feasibility of SHAVE across a range of configurations. We now turn to a complementary exercise that examines the sufficiency of the constructed set $\mathcal{V}$ in generating sign changes. To this end, we focus on a least favorable regime suggested by Table 2, namely, settings with larger sample size and signal strength and weaker dependence.

Specifically, we consider $n \in \{30, 100\}$ and $(\beta_1, \beta_2) \in \{(0.2, 0.2), (1.0, 1.0)\}$, with error variance fixed at $\sigma^2 = 0.5$ in the data-generating process (4.1). The regressors $X_1$ and $X_2$ are generated independently from $N(0, I_n)$. The auxiliary variables in $X_A$ are constructed according to

$$V = X_1 A_1 + X_2 A_2 + bY + \epsilon \tilde{U}, \tag{4.4}$$

where $\tilde{U} = U/\|U\|$ with $U \sim N(0, I_n)$, $(A_1, A_2) \in \{(0.1, -0.1), (1.0, -1.0)\}$ and $b \in \{0.1, 1.0\}$. The perturbation level follows $\epsilon \sim Unif([0, \xi])$, with $\xi \in \{0.1, 1.0\}$. For each of 100 Monte Carlo replications, $10^4$ auxiliary variables are generated.

Building on the previous simulation, we further examine whether the sufficient conditions underlying the construction of $\mathcal{V}$ are satisfied, while maintaining the same evaluation metrics. Specifically, we consider (C1) $0 < \varepsilon \leq \eta \|M_{X1,X2}Y\| \kappa_{\min}(A)$ and (C2) $b \geq \kappa_{\max}(A)$, where $\kappa_{\min}(A)$ and $\kappa_{\max}(A)$ are defined in Eq. (2.2). We set $\eta = 0.24$.

Table 3 summarizes the results. When both conditions (C1) and (C2) are satisfied, SHAVE occurs with probability one, and the resulting estimates are statistically significant, providing direct empirical support for the sufficiency of $\mathcal{V}$. Even when these conditions are not jointly satisfied, SHAVE continues to occur with high probability when the perturbation level is small ($\xi = 0.1$). For larger perturbations ($\xi = 1$), conditions (C1) and (C2) are rarely satisfied simultaneously. Nevertheless, SHAVE still occurs at a high rate, often accompanied by statistically significant estimates.

These findings highlight that the constructed set $\mathcal{V}$ captures a tractable sufficient subset of sign-reversing perturbations, but does not exhaust all such directions. The observed patterns are driven by the construction parameters and the data-generating process: stronger dependence between $V$ and $(X_1, X_2, Y)$ increases the likelihood of sign reversal, whereas larger perturbations or stronger signals tend to move $V$ away from regions that induce sign changes.

The first and second simulations together verify the existence of a set of auxiliary variables with positive Lebesgue measure that induce SHAVE, although identifying such variables becomes increasingly difficult under conditional independence. The constructed set $\mathcal{V}$ provides a sufficient but not necessary characterization: realizations outside $\mathcal{V}$ may still induce SHAVE, and the construction does not fully capture settings with highly correlated covariates.

Importantly, the variables in $X_A$ have no causal effect on $Y$ under either the data-generating process of $Y$ or the construction of $V$. All observed sign reversals arise entirely from sample-induced dependence between $(X_1, X_2, Y)$ and $V$, rather than from genuine causal relationships. From the perspective of the hacker, this suggests that the availability of variables sufficiently correlated with the existing covariates facilitates the implementation of SHAVE.

A further complication is that baseline estimates may themselves exhibit sign uncertainty due to sampling variability, as reflected in column "Valid", a phenomenon that is often exacerbated by strong correlation among covariates. In such cases, although the inclusion of an auxiliary variable can be viewed as deliberate manipulation, the resulting sign reversal may coincidentally align the estimate with the true direction. In other words, SHAVE, while fundamentally driven by spurious dependence, may sometimes appear to "correct" an initially incorrect sign. This dual possibility, in which SHAVE can either distort or seemingly improve inference, makes it inherently difficult to distinguish from genuine signal, and may even render the two observationally indistinguishable. Motivated by this challenge, we develop diagnostic tools to detect potential SHAVE in Section 5.

## 4.3   An empirical example using the rat eye gene dataset

We next examine how SHAVE may manifest in empirical analysis using the rat eye gene expression dataset (Scheetz et al., 2006). The data consist of microarray measurements from eye tissue of 120 twelve-week-old male rats, obtained using the Affymetrix Rat Genome 230 2.0 Array with 31,099 probe sets. Following preprocessing for expression level and variability (Huang

Table 3: Frequency of SHAVE under constructed auxiliary variables.

| $n$ | $(\beta_1,\beta_2)$ | $(A_1,A_2)$ | $b$ | Valid | C1 | C2 | SHAVE | Sig.$(\hat{\alpha}_1)$ | Sig.$(\hat{\alpha}_2)$ |
|---|---|---|---|---|---|---|---|---|---|
| Panel A. $\xi = 0.1$ | | | | | | | | | |
| 30 | (0.2,0.2) | (0.1,-0.1) | 0.1 | 83.0% | 81.9% | 1.2% | 100.0% | 100.0% | 100.0% |
| | | | 1 | 88.0% | 80.7% | 100.0% | 100.0% | 100.0% | 100.0% |
| | | (1,-1) | 0.1 | 88.0% | 100.0% | 0.0% | 100.0% | 100.0% | 100.0% |
| | | | 1 | 87.0% | 100.0% | 1.1% | 100.0% | 100.0% | 100.0% |
| | (1,1) | (0.1,-0.1) | 0.1 | 100.0% | 0.0% | 95.0% | 100.0% | 100.0% | 100.0% |
| | | | 1 | 100.0% | 0.0% | 100.0% | 100.0% | 100.0% | 100.0% |
| | | (1,-1) | 0.1 | 100.0% | 100.0% | 0.0% | 100.0% | 100.0% | 100.0% |
| | | | 1 | 100.0% | 100.0% | 94.0% | 100.0% | 100.0% | 100.0% |
| 100 | (0.2,0.2) | (0.1,-0.1) | 0.1 | 100.0% | 100.0% | 0.0% | 100.0% | 100.0% | 100.0% |
| | | | 1 | 100.0% | 100.0% | 100.0% | 100.0% | 100.0% | 100.0% |
| | | (1,-1) | 0.1 | 100.0% | 100.0% | 0.0% | 100.0% | 100.0% | 100.0% |
| | | | 1 | 98.0% | 100.0% | 0.0% | 100.0% | 100.0% | 100.0% |
| | (1,1) | (0.1,-0.1) | 0.1 | 100.0% | 100.0% | 99.0% | 100.0% | 100.0% | 100.0% |
| | | | 1 | 100.0% | 100.0% | 100.0% | 100.0% | 100.0% | 100.0% |
| | | (1,-1) | 0.1 | 100.0% | 100.0% | 0.0% | 100.0% | 100.0% | 100.0% |
| | | | 1 | 100.0% | 100.0% | 100.0% | 100.0% | 100.0% | 100.0% |
| Panel B. $\xi = 1.0$ | | | | | | | | | |
| 30 | (0.2,0.2) | (0.1,-0.1) | 0.1 | 86.0% | 0.0% | 0.0% | 77.6% | 62.2% | 97.8% |
| | | | 1 | 88.0% | 0.0% | 100.0% | 99.7% | 93.2% | 98.7% |
| | | (1,-1) | 0.1 | 92.0% | 87.0% | 0.0% | 99.4% | 88.2% | 93.9% |
| | | | 1 | 90.0% | 80.0% | 0.0% | 100.0% | 100.0% | 100.0% |
| | (1,1) | (0.1,-0.1) | 0.1 | 100.0% | 0.0% | 95.0% | 39.4% | 70.7% | 100.0% |
| | | | 1 | 100.0% | 0.0% | 100.0% | 94.1% | 71.5% | 95.8% |
| | | (1,-1) | 0.1 | 100.0% | 0.0% | 0.0% | 96.2% | 75.6% | 100.0% |
| | | | 1 | 100.0% | 0.0% | 93.0% | 100.0% | 100.0% | 100.0% |
| 100 | (0.2,0.2) | (0.1,-0.1) | 0.1 | 100.0% | 0.0% | 0.0% | 100.0% | 98.8% | 100.0% |
| | | | 1 | 100.0% | 0.0% | 100.0% | 100.0% | 100.0% | 100.0% |
| | | (1,-1) | 0.1 | 100.0% | 100.0% | 0.0% | 100.0% | 100.0% | 100.0% |
| | | | 1 | 99.0% | 100.0% | 0.0% | 100.0% | 100.0% | 100.0% |
| | (1,1) | (0.1,-0.1) | 0.1 | 100.0% | 0.0% | 100.0% | 71.3% | 83.1% | 100.0% |
| | | | 1 | 100.0% | 0.0% | 100.0% | 100.0% | 100.0% | 100.0% |
| | | (1,-1) | 0.1 | 100.0% | 100.0% | 0.0% | 100.0% | 100.0% | 100.0% |
| | | | 1 | 100.0% | 100.0% | 100.0% | 100.0% | 100.0% | 100.0% |

Note: "C1" and "C2" denote the frequencies with which conditions C1 and C2 hold. The remaining columns are defined as in Table 2. Entries are percentages based on $10^4$ generated realizations of constructed auxiliary variables across 100 Monte Carlo replications.

et al., 2008), 19,032 probes are retained.

Previous studies have identified at most 24 probes that may be significantly associated with the TRIM32 gene (Huang et al., 2008). However, even when restricting attention to the top 200 probes ranked by marginal correlation, commonly used selection methods such as Lasso, SCAD, and MCP exhibit varying degrees of instability (Nan and Yang, 2014). From the perspective of a sign hacker, such instability creates scope for implementing SHAVE, as it reflects the sensitivity

of estimation outcomes to model specification, thereby allowing sign changes to arise under suitable augmentation by auxiliary variables.

Suppose the hacker also sets probe 1389163_at as the response variable $Y$ and constructs a baseline regression of $Y$ on $X_{\mathcal{I}_1}$ and $X_{\mathcal{C}}$, where both are drawn from the 24 TRIM32-associated probes. Here, $X_{\mathcal{I}_1}$ contains the covariate(s) of primary interest, while $X_{\mathcal{C}}$ serves as the control set. Table 4 reports the regression results for the two scenarios described below, comparing outcomes before and after SHAVE.

Table 4: Sign change phenomenon for $X_{\mathcal{I}_1}$

| Cat. | Var. | M1 | M2 | M3 | M4 |
|------|------|-----|-----|-----|-----|
| $X_{\mathcal{I}_1}$ | 1369353_at | -0.018 | 0.214* | -0.142 | $-0.163^*$ |
| | | (0.119) | (0.128) | (0.102) | (0.095) |
| | 1370429_at | – | – | -0.010 | 0.193* |
| | | | | (0.109) | (0.113) |
| $V$ | 1372894_at | – | 0.278*** | – | – |
| | | | (0.073) | | |
| | 1375833_at | – | – | – | $-0.310^{***}$ |
| | | | | | (0.075) |
| $X_{\mathcal{C}}$ | 1371242_at | -0.081 | -0.053 | – | – |
| | | (0.115) | (0.108) | | |
| | 1374106_at | 0.092 | 0.172* | 0.229*** | 0.235*** |
| | | (0.093) | (0.090) | (0.078) | (0.073) |
| | 1374131_at | 0.096 | 0.077 | 0.108 | 0.139* |
| | | (0.075) | (0.070) | (0.076) | (0.071) |
| | 1378935_at | -0.082 | -0.022 | -0.101 | -0.081 |
| | | (0.094) | (0.090) | (0.089) | (0.083) |
| | 1380033_at | 0.178** | 0.178*** | – | – |
| | | (0.069) | (0.065) | | |
| | 1382835_at | – | – | 0.213*** | 0.147** |
| | | | | (0.065) | (0.063) |
| | 1383110_at | 0.189 | 0.255** | – | – |
| | | (0.122) | (0.116) | | |
| | 1383522_at | 0.081 | 0.122 | 0.103 | 0.065 |
| | | (0.082) | (0.078) | (0.076) | (0.072) |
| | 1383673_at | -0.013 | -0.042 | – | – |
| | | (0.117) | (0.110) | | |
| | 1383749_at | -0.102 | $-0.150^{**}$ | -0.111 | $-0.120^*$ |
| | | (0.074) | (0.071) | (0.068) | (0.063) |
| | 1383996_at | – | – | 0.213*** | 0.207*** |
| | | | | (0.060) | (0.056) |
| | 1389584_at | 0.329*** | 0.306*** | – | – |
| | | (0.107) | (0.101) | | |
| | 1390788_a_at | 0.130 | 0.199** | – | – |
| | | (0.089) | (0.086) | | |
| | 1393382_at | 0.043 | 0.044 | – | – |
| | | (0.082) | (0.077) | | |
| | 1393684_at | 0.042 | 0.028 | 0.115 | 0.107 |
| | | (0.082) | (0.077) | (0.073) | (0.068) |
| | 1393979_at | -0.044 | -0.107 | – | – |
| | | (0.124) | (0.118) | | |
| | 1394107_at | – | – | $-0.299^{***}$ | $-0.314^{***}$ |
| | | | | (0.092) | (0.086) |
| | 1395415_at | -0.008 | 0.011 | 0.144* | 0.151* |
| | | (0.097) | (0.091) | (0.087) | (0.081) |
| | 1398255_at | $-0.366^{***}$ | $-0.337^{***}$ | – | – |
| | | (0.095) | (0.090) | | |
| BIC | | 260.26 | 249.2 | 234.71 | 221.5 |

Notes: Standard errors are reported in parentheses. ***$p < 0.01$, **$p < 0.05$, *$p < 0.1$. $|\mathcal{C}|$ denotes the number of controls.

In the first scenario, $X_{\mathcal{I}_1}$ consists of a single probe (1369353_at) and $X_{\mathcal{C}}$ contains 16 controls. The baseline regression yields a negative coefficient for 1369353_at, which supposedly does not align with the hacker's desired narrative. The hacker then searches over the remaining 19,007 probes and identifies an auxiliary variable $V$ (1372894_at) such that, upon inclusion, the coefficient of 1369353_at becomes positive and statistically significant at the 10% level.

In the second scenario, $X_{\mathcal{I}_1}$ contains two probes (1369353_at and 1370429_at) and $X_{\mathcal{C}}$ contains 10 controls. The baseline regression produces negative coefficients for both variables, whereas the hacker's desired pattern is a significantly negative effect for 1369353_at and a significantly positive effect for 1370429_at. Again, by searching over the remaining probes, the hacker successfully identifies an auxiliary variable $V$ (1375833_at) that induces this selective sign reversal.

In both scenarios, the selected auxiliary variable $V$ is itself statistically significant, while the estimated coefficients of the control variables remain largely unchanged. From the perspective of standard model diagnostics, the manipulated specifications appear preferable: models including $V$ consistently exhibit lower BIC values than those excluding it, and $F$-tests comparing the two specifications reject the null hypothesis that the coefficient on $V$ is zero. These diagnostic outcomes provide conventional statistical justification for including $V$, reinforcing the apparent credibility of the resulting model. These results demonstrate that SHAVE is practically implementable and yields empirical consequences consistent with the theoretical analysis in Section 3.

The two scenarios presented above are representative rather than exceptional. Additional analyses reveal that, for the same $Y$ and $X_{\mathcal{I}_1}$ as in the above setup, each control set size gives rise to many distinct specifications, within which multiple auxiliary variables $V$ can be identified to induce the desired sign pattern. Their number declines mildly as control set size increases. Moreover, even when the baseline model includes all 24 TRIM32-associated probes, searching over the remaining 19,007 probes continues to yield a large number of candidate variables capable of inducing sign reversals in one or more target coefficients, often in the thousands. Importantly, the resulting estimates remain statistically significant, and model comparisons continue to favor specifications that include the selected auxiliary variable, even as the control set varies. These additional findings provide empirical evidence that SHAVE is not only feasible but also easy to implement and can readily arise in practice. Further details are provided in Appendix D. Since the true data-generating process is never observable, the baseline model without $V$ cannot be regarded as inherently more reliable. As a result, empirical evidence alone may not distinguish between baseline and augmented models, even when they imply markedly different sign patterns.

This practical feasibility can be further reinforced by the availability of standard statistical software. For instance, the *tuples* command in Stata can enumerate variable combinations [3], while the *FWDselect* package in R automates model search based on criteria such as $R^2$ or residual variance. Although these tools are designed for legitimate exploratory analysis, they can also be misused to search over the response $Y$, the variable of interest $X$, and a large auxiliary pool $X_A$ for specifications that improve in-sample fit and deliver a preferred sign pattern. In this sense, the practical feasibility of SHAVE reflects not only the abundance of candidate auxiliary variables in modern data settings, but also the ease with which large-scale specification search can be operationalized.

---

[3]The *tuples* command functions efficiently for fewer than 20 variables, but may exceed memory capacity for larger sets.

# 5 Detecting SHAVE

The model produced by SHAVE often appears statistically valid when evaluated on the reported data, and the results are even reproducible within that dataset. This makes distinguishing genuine findings from manipulated specifications particularly challenging. We propose two detection strategies that exploit independent information: (i) lineup detection using augmented data and (ii) replication detection using an independent dataset or a new experimental design.

## 5.1 Lineup detection using augmented data

Suppose we suspect that including a variable $V$ induces SHAVE for a target variable $X_{\mathcal{I}_1}$ in a regression model. We examine how often $V$ is selected compared with other variables of similar characteristics, based on prior studies or expert knowledge. If $V$ genuinely contributes to the outcome $Y$, it should be selected more frequently than comparable variables without direct influence. A low selection frequency instead suggests weak empirical support, raising concerns about potential SHAVE.

If $V$ is rarely selected because it is highly correlated with newly included variables, the conclusion remains inconclusive, as these correlated substitutes may serve as proxies for $V$. In this case, we reestimate the model by replacing $V$ with its correlated counterparts and examine whether the estimated signs for $X_{\mathcal{I}_1}$ remain stable. Stability of the signs provides indirect support for the role of $V$, whereas sign reversals suggest possible manipulation. This procedure, analogous to a police lineup, is termed lineup detection.

We now illustrate the performance of the lineup detection procedure applied to the results after SHAVE in Section 4.3. For each identified candidate $V$, we augment the set of remaining genes and examine its selection frequency under Lasso, SCAD, and MCP with cross-validated tuning parameters, both with and without conditioning on the inclusion of $X_{\mathcal{I}_1}$. The procedure is repeated 50 times to account for selection instability. Under the same setup of $Y$ and $X_{\mathcal{I}_1}$, none of the identified auxiliary variables $V$ are selected by these methods in either the conditional or unconditional specifications. Extending the analysis across all probes in the 24 TRIM32-associated genes yields similar patterns: after accounting for selection instability, only a small fraction (typically below 1%) of candidate variables are selected by SCAD and MCP, with slightly higher frequencies observed for Lasso and under conditional selection . Additional numerical simulations reinforce this pattern: variables that are spuriously associated with the outcome tend to have low selection frequencies even when correlated with true predictors, whereas genuinely relevant variables are selected consistently more often. See Appendix E for further details. Taken together, these findings suggest that lineup detection can provide useful diagnostic information for distinguishing manipulated specifications, although it does not offer a definitive identification criterion.

## 5.2 Replication detection with an independent dataset

Replicating results using independent data provides a more direct diagnostic for SHAVE. Let the suspected model yield estimate $\hat{\beta}_{X,j}$ with standard error $se(\hat{\beta}_{X,j})$ and sample size $n_0$, together with the $(1 - \alpha_0)$-level confidence interval $[\hat{L}_j, \hat{U}_j]$. If $[\hat{L}_j, \hat{U}_j]$ excludes zero, say $\hat{L}_j > 0$, the original claim is that $\beta_j > 0$.

### 5.2.1 Detection with another independent dataset

Suppose an independent dataset of size $n_1$ is available. Let $\tilde{\beta}_{X,j}$ be the new estimate, with adjusted standard error $se(\tilde{\beta}_{X,j}) = \sqrt{\frac{n_0}{n_1}} se(\hat{\beta}_{X,j})$ to ensure comparable scaling.

We test $H_0 : \beta_{X,j} \geq \tau \hat{\beta}_{X,j}$ versus $H_1 : \beta_{X,j} < \tau \hat{\beta}_{X,j}$, where $\tau \in (0,1)$ is a believability index reflecting the replicator's confidence in the original claim. Failure to reject $H_0$ provides support for the original finding, whereas rejection indicates inconsistency that may arise from potential SHAVE.

If no prior belief about $\tau$ exists, a data-driven benchmark $\tau^* = \frac{\tilde{\beta}_{X,j} + c_{\alpha_0} se(\tilde{\beta}_{X,j})}{\hat{\beta}_{X,j}}$ can be computed, where $c_{\alpha_0}$ is the one-sided critical value of the $t$-distribution at level $\alpha_0$. Smaller values of $\tau^*$ indicate greater inconsistency between the two studies. When $\tau^* \leq \hat{L}_j / \hat{\beta}_{X,j}$, the confidence intervals for $\tilde{\beta}_{X,j}$ and $\hat{\beta}_{X,j}$ do not overlap, providing strong evidence against the original specification.

### 5.2.2 Detection via new study design

When independent data are unavailable, replication can be planned through a new study with sufficient statistical power.

To assess the suspected result $\hat{\beta}_{X,j} > 0$, we test $H_0 : \beta_{X,j} \geq \tau_1\hat{\beta}_{X,j}$ against $H_1 : \beta_{X,j} < \tau_1\hat{\beta}_{X,j}$ with predetermined believability index $\tau_1 \in (0,1)$, significance level $\alpha_1$, and type II error rate $\zeta_1$. The required sample size is $n_2' = n_0 se(\hat{\beta}_{X,j})^2 \left(\frac{c_{\zeta_1}+c_{\alpha_1}}{\tau_1\hat{\beta}_{X,j}}\right)^2$. Conversely, to confirm the claim, we test $H_0 : \beta_{X,j} \leq 0$ versus $H_1 : \beta_{X,j} > 0$. Ensuring power of at least $1 - \zeta_2$ under the alternative $\beta_{X,j} = \tau_2\hat{\beta}_{X,j}$ requires $n_2'' = n_0 se(\hat{\beta}_{X,j})^2 \left(\frac{c_{\zeta_2}+c_{\alpha_2}}{\tau_2\hat{\beta}_{X,j}}\right)^2$, where $\alpha_2$ and $\tau_2$ denote the type I error and believability index for confirmation. A balanced design with $n_2 \geq \max(n_2', n_2'')$ ensures sufficient power to either refute or confirm the suspected finding, while maintaining type II error below $\min(\zeta_1, \zeta_2)$. The same logic applies when the suspected coefficient is negative.

Although SHAVE can be statistically detected, its prevention remains challenging due to limited information and the high cost of collecting new data. Detection procedures therefore play a critical role in assessing the credibility of empirical findings, particularly when replication is infeasible. More broadly, these approaches highlight the importance of incorporating external or independent information when evaluating potentially manipulated specifications.

## 6  Conclusion

In this paper, we investigate the possibility of Sign Hacking with Auxiliary Variable Exploration (SHAVE) in the context of big data. SHAVE refers to the introduction of auxiliary variables to achieve a desired sign pattern of estimated coefficients. By representing variables as vectors in Euclidean space, we prove the existence of a set of auxiliary variables with positive Lebesgue measure whose inclusion can reverse the signs of existing coefficients. We further show that, with high probability, such variables can be identified from a large pool of candidates, and that both $t$-statistics and standard measures of model fit can be arbitrarily manipulated. Together, these results provide a formal mechanism through which specification search can generate statistically significant yet potentially spurious findings. Numerical examples further indicate that strong dependence among covariates increases susceptibility to such manipulation.

These findings have important implications for empirical practice. In settings with substantial model uncertainty, particularly in high-dimensional applications, selective inclusion of auxiliary variables can produce results that appear statistically valid and internally consistent, while deviating from the underlying data generating mechanism. Because the manipulated specifications may remain reproducible within the observed dataset, conventional diagnostic tools offer limited protection against such behavior. This highlights the role of model uncertainty as a fundamental source of vulnerability in empirical analysis.

We propose detection strategies based on augmented or independent data to assess the credibility of such findings. While these procedures can help distinguish between spurious and genuinely relevant variables, their effectiveness is inherently limited when external information is unavailable. Moreover, our analysis focuses on the inclusion of a single auxiliary variable, whereas in practice multiple variables may jointly induce similar effects. Extending these strategies to more complex settings, including multi-variable and nonlinear specifications, remains an important direction for future research.

More broadly, SHAVE underscores a fundamental limitation of inference based solely on correlation. Modern empirical practice often relies on linear models in which coefficients are interpreted as causal effects. Yet, as Fisher (1950, p. 190) cautioned, "If we choose ... a group of social phenomena with no antecedent knowledge of the causation or absence of causation among them, then the calculation of correlation coefficients, total or partial, will not advance us a step towards evaluating the importance of the causes at work." This observation remains highly relevant: statistical significance in the presence of flexible specification search may reflect artefacts of model construction rather than genuine causal relationships.

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

# Appendix

The appendix is organized as follows. Section A collects the proofs of the main theoretical results: Section A.1 proves the geometric sign-change results in Section 2, and Section A.2 proves the SHAVE results in Section 3, including the high-probability search result and the results for *t*- and *F*-statistics. Section B provides supplementary geometric characterizations of sign reversal in both planar and non-planar settings. Section C discusses connections between SHAVE and related empirical phenomena, including spurious correlation, p-hacking, sponsorship bias, and research misconduct. Sections D–F report additional empirical and simulation results, including auxiliary-variable identification, lineup detection, and details of the body fat dataset.

# A   Proofs of main results

## A.1   Proof of results in Section 2

***Proof of Theorem 2.1.*** It suffices to verify the claim for the constructed set $\mathcal{V}$, since one may take $\mathcal{V}^* = \mathcal{V}$. Specifically, we only need to show that every $\vec{v} \in \mathcal{V}$ satisfies the sign relation in (2.1) upon inclusion, and that $\mathcal{V}$ is measurable with positive Lebesgue measure.

For each $j$, partial out $\boldsymbol{X}_{-j}$ and define $\tilde{y} = M_{\boldsymbol{X}_{-j}}\vec{y}$, $\tilde{\boldsymbol{X}}_j = M_{\boldsymbol{X}_{-j}}\boldsymbol{X}_j$, $\tilde{v} = M_{\boldsymbol{X}_{-j}}\vec{v}$ and $\tilde{u} = M_{\boldsymbol{X}_{-j}}\vec{u}$. By the Frisch-Waugh-Lovell theorem,

$$\hat{\beta}_{\boldsymbol{X},j} = \frac{\tilde{\boldsymbol{X}}_j'\tilde{y}}{\tilde{\boldsymbol{X}}_j'\tilde{\boldsymbol{X}}_j}, \quad \hat{\alpha}_v = \frac{\tilde{v}'M_{\tilde{\boldsymbol{X}}_j}\tilde{y}}{\tilde{v}'M_{\tilde{\boldsymbol{X}}_j}\tilde{v}}, \quad \hat{\alpha}_{\boldsymbol{X},j} = \hat{\beta}_{\boldsymbol{X},j} - \frac{\tilde{\boldsymbol{X}}_j'\tilde{v}}{\tilde{\boldsymbol{X}}_j'\tilde{\boldsymbol{X}}_j}\hat{\alpha}_v.$$

Since $\tilde{\boldsymbol{X}}_j$ lies in the orthogonal complement of $\boldsymbol{X}_{-j}$, $M_{\tilde{\boldsymbol{X}}_j} M_{\boldsymbol{X}_{-j}} = M_{\boldsymbol{X}}$, implying $T_2 = \tilde{y}' M_{\tilde{\boldsymbol{X}}_j} \tilde{y} = \|M_{\boldsymbol{X}} \vec{y}\|^2$. Write $\vec{v} = \boldsymbol{X} A + b\vec{y} + \varepsilon \vec{u}$. Then

$$\tilde{v} = \tilde{\boldsymbol{X}}_j A_j + b\tilde{y} + \varepsilon \tilde{u}, \quad M_{\tilde{\boldsymbol{X}}_j} \tilde{v} = b M_{\tilde{\boldsymbol{X}}_j} \tilde{y} + \varepsilon M_{\tilde{\boldsymbol{X}}_j} \tilde{u}.$$

Let

$$T_1 = (\tilde{\boldsymbol{X}}_j' \tilde{\boldsymbol{X}}_j)^{-1} \tilde{\boldsymbol{X}}_j' \tilde{u}, \quad T_3 = \tilde{y}' M_{\tilde{\boldsymbol{X}}_j} \tilde{u}, \quad T_4 = \tilde{u}' M_{\tilde{\boldsymbol{X}}_j} \tilde{u}.$$

Straightforward algebra yields

$$(\tilde{\boldsymbol{X}}_j' \tilde{\boldsymbol{X}}_j)^{-1} \tilde{\boldsymbol{X}}_j' \tilde{v} \hat{\alpha}_v = \frac{A_j b T_2 + b^2 \hat{\beta}_{\boldsymbol{X},j} T_2 + (A_j T_3 + b T_3 \hat{\beta}_{\boldsymbol{X},j} + b T_1 T_2)\varepsilon + T_1 T_3 \varepsilon^2}{b^2 T_2 + 2b T_3 \varepsilon + \varepsilon^2 T_4}.$$

Using $\hat{\alpha}_{\boldsymbol{X},j} = \hat{\beta}_{\boldsymbol{X},j} - (\tilde{\boldsymbol{X}}_j' \tilde{\boldsymbol{X}}_j)^{-1} \tilde{\boldsymbol{X}}_j' \tilde{v} \hat{\alpha}_v$, a direct calculation gives

$$\hat{\alpha}_{\boldsymbol{X},j} = \frac{-A_j b T_2 + \left[ T_3(b\hat{\beta}_{\boldsymbol{X},j} - A_j) - b T_1 T_2 \right]\varepsilon + \left[ \hat{\beta}_{\boldsymbol{X},j} T_4 - T_1 T_3 \right]\varepsilon^2}{b^2 T_2 + 2b T_3 \varepsilon + \varepsilon^2 T_4}.$$

We next bound the perturbation terms in the numerator. Since $\|\vec{u}\| = 1$, we have $\|\tilde{u}\| \leq 1$, hence

$$|T_1| \leq \|\tilde{\boldsymbol{X}}_j\|^{-1}, \qquad |T_3| \leq \|M_{\boldsymbol{X}} \vec{y}\|, \qquad T_4 \leq 1, \qquad |\hat{\beta}_{\boldsymbol{X},j}| \leq \frac{\|\tilde{y}\|}{\|\tilde{\boldsymbol{X}}_j\|}.$$

Also, by the definitions of $\kappa_{\min}(A)$ and $\kappa_{\max}(A)$, $\varepsilon \leq \eta \|M_{\boldsymbol{X}} \vec{y}\| \kappa_{\min}(A) \leq \eta \|M_{\boldsymbol{X}} \vec{y}\| \frac{|A_j| \|\tilde{\boldsymbol{X}}_j\|}{\|\tilde{y}\|}$, and $b \geq \kappa_{\max}(A) \geq \frac{|A_j| \|\tilde{\boldsymbol{X}}_j\|}{\|\tilde{y}\|}$. Using these bounds, together with $\|M_{\boldsymbol{X}} \vec{y}\| \leq \|\tilde{y}\|$, we obtain

$$|A_j T_3 \varepsilon| \leq |A_j| \|M_{\boldsymbol{X}} \vec{y}\| \varepsilon \leq \eta |A_j| b T_2,$$

$$|b T_3 \hat{\beta}_{\boldsymbol{X},j} \varepsilon| \leq b \|M_{\boldsymbol{X}} \vec{y}\| \frac{\|\tilde{y}\|}{\|\tilde{\boldsymbol{X}}_j\|} \varepsilon \leq \eta |A_j| b T_2,$$

$$|b T_1 T_2 \varepsilon| \leq b \|\tilde{\boldsymbol{X}}_j\|^{-1} T_2 \varepsilon \leq \eta |A_j| b T_2,$$

$$|\hat{\beta}_{\boldsymbol{X},j} T_4 \varepsilon^2| \leq \frac{\|\tilde{y}\|}{\|\tilde{\boldsymbol{X}}_j\|} \varepsilon^2 \leq \eta^2 |A_j| b T_2,$$

$$|T_1 T_3 \varepsilon^2| \leq \|\tilde{\boldsymbol{X}}_j\|^{-1} \|M_{\boldsymbol{X}} \vec{y}\| \varepsilon^2 \leq \eta^2 |A_j| b T_2.$$

Therefore, the numerator of $\hat{\alpha}_{\boldsymbol{X},j}$ differs from $-A_j b T_2$ by at most $(3\eta + 2\eta^2)|A_j| b T_2$. Since $\eta \in (0, 1/4)$, we have $3\eta + 2\eta^2 < 1$. Hence the numerator has the same sign as $-A_j b T_2$. Moreover,

$$b^2 T_2 + 2b T_3 \varepsilon + \varepsilon^2 T_4 \geq b^2 T_2 - 2b |T_3| \varepsilon \geq b^2 T_2 (1 - 2\eta) > 0,$$

so the denominator is strictly positive. Therefore,

$$\operatorname{sgn} \hat{\alpha}_{\boldsymbol{X},j} = \operatorname{sgn}(-A_j/b).$$

Under the normalization $b > 0$ imposed in the construction of $\mathcal{V}$, $A_j \hat{\beta}_{\boldsymbol{X},j} > 0$ for $j \in \mathcal{I}_1$ and $A_j \hat{\beta}_{\boldsymbol{X},j} < 0$ for $j \in \mathcal{I}_2$, implying

$$\operatorname{sgn} \hat{\beta}_{\boldsymbol{X},\mathcal{I}_1} = -\operatorname{sgn} \hat{\alpha}_{\boldsymbol{X},\mathcal{I}_1}, \quad \operatorname{sgn} \hat{\beta}_{\boldsymbol{X},\mathcal{I}_2} = \operatorname{sgn} \hat{\alpha}_{\boldsymbol{X},\mathcal{I}_2}.$$

It remains to show that $\mathcal{V}$ is measurable and has positive Lebesgue measure. Since $b$ is fixed in the construction, we can write $\mathcal{V} = \{\boldsymbol{X} A + b\vec{y} + w : 0 < \|w\| \leq \varepsilon_0\}$. Thus $\mathcal{V}$ is a translation of the punctured closed ball $\{w \in \mathbb{R}^n : 0 < \|w\| \leq \varepsilon_0\}$, and hence is Lebesgue measurable. Fix any $u_0 \in \mathbb{S}^{n-1}$. Set $w_0 = \varepsilon_0 u_0/2$ and $v_1 = \boldsymbol{X} A + b\vec{y} + w_0$. Then $v_1 \in \mathcal{V}$. Moreover, if $\|h\| < \varepsilon_0/4$, then $\varepsilon_0/4 < \|w_0 + h\| < 3\varepsilon_0/4$, so that $v_1 + h \in \mathcal{V}$.

Hence $B(v_1, \varepsilon_0/4) \subseteq \mathcal{V}$, which implies $\mu(\mathcal{V}) \geq \mu(B(v_1, \varepsilon_0/4)) > 0$. Therefore $\mathcal{V}$ is a measurable set with positive Lebesgue measure, and taking $\mathcal{V}^* = \mathcal{V}$ completes the proof. ∎

## A.2  Proof of results in Section 3

**Lemma A.1.** *Let $X$ be a real random variable with density $f_X$ satisfying $f_X(x) \geq c > 0$ for all $x \in [-R, R]$. Let $X_1, \ldots, X_n$ be i.i.d. copies of $X$ and $\tilde{X} = (X_1, \ldots, X_n)$. For any $\tilde{x} \in \mathbb{R}^n$ with $\|\tilde{x}\|_2 = r < R$ and any $0 < \epsilon < R - r$, we have*

$$P(\tilde{X} \in B_2(\tilde{x}, \epsilon)) > 0.$$

*where $B_2(\tilde{x}, \epsilon)$ is an open ball.*

**Proof of Lemma A.1.** Write $\tilde{x} = (\tilde{x}_1, \ldots, \tilde{x}_n)$. Because $\|\tilde{x}\|_2 = r < R$, we have $|\tilde{x}_i| < R$ for each $i$. Choose $\delta = \epsilon/(2\sqrt{n})$. Then for each $i$, set $I_i = (\tilde{x}_i - \delta, \tilde{x}_i + \delta) \subset (-R, R)$. Because $|\tilde{x}_i| \leq r$ and $\delta = \epsilon/(2\sqrt{n}) < \epsilon < R - r$, we have $I_i \subset (-R, R)$. Hence

$$P(X_i \in I_i) = \int_{I_i} f_X(x)dx \geq 2c\delta > 0.$$

By independence,

$$P\left(\cap_{i=1}^n \{X_i \in I_i\}\right) = \prod_{i=1}^n P(X_i \in I_i) \geq (2c\delta)^n > 0.$$

On the event $\cap_{i=1}^n \{X_i \in I_i\}$,

$$\|\tilde{X} - \tilde{x}\|^2 = \sum_{i=1}^n (X_i - \tilde{x}_i)^2 < \sum_{i=1}^n \delta^2 = n\delta^2 = \left(\frac{\epsilon}{2}\right)^2,$$

so $\tilde{X} \in B_2(\tilde{x}, \epsilon)$, where $B_2(\tilde{x}, \epsilon)$ denotes the open Euclidean ball centered at $\tilde{x}$ with radius $\epsilon$. Therefore,

$$P\left(\tilde{X} \in B_2(\tilde{x}, \epsilon)\right) \geq (2c\delta)^n > 0. \qquad ∎$$

**Proof of Theorem 3.1.** Let $\mathcal{V}$ be defined in (2.4) under constraints (2.3). By Theorem 2.1, the set $\mathcal{V}$ is measurable with positive Lebesgue measure, and every $v \in \mathcal{V}$ satisfies the sign relation in (3.3).

Define the spherical success set

$$\mathcal{W} = \left\{\omega \in \mathbb{S}^{n-1} : \int_0^\infty \mathbb{1}_{\mathcal{V}}(r\omega)r^{n-1}dr > 0\right\}.$$

Applying polar coordinates directly to $\mathcal{V}$,

$$\mu(\mathcal{V}) = \int_{\mathbb{S}^{n-1}} \left(\int_0^\infty \mathbb{1}_{\mathcal{V}}(r\omega)r^{n-1}dr\right) d\tau(\omega),$$

where $\tau(\cdot)$ denotes the Hausdorff measure on the unit sphere. If $\tau(\mathcal{W}) = 0$, then $\int_0^\infty \mathbb{1}_{\mathcal{V}}(r\omega)r^{n-1}dr = 0$ for $\tau$-a.e. $\omega \in \mathbb{S}^{n-1}$, which implies $\mu(\mathcal{V}) = 0$, contradicting $\mu(\mathcal{V}) > 0$. Hence $\tau(\mathcal{W}) > 0$.

Let $\{V^{(j)}\}_{j \in \mathcal{J}} \subseteq X_A$ be the conditionally independent variables in Assumption 1 and write their normalized realizations as $v^{(j)} = \frac{V^{(j)}}{\|V^{(j)}\|} \in \mathbb{S}^{n-1}$. Since $v^{(j)}$ is a measurable transformation of $V^{(j)}$, the variables $\{v^{(j)}\}_{j \in \mathcal{J}}$ remain conditionally independent given $(Y, X)$. By Assumption 1, there exists a constant $c_0 > 0$ such that, for every $j \in \mathcal{J}$, the conditional density $f_j(\cdot)$ of $v^{(j)}$ w.r.t. $\tau(\cdot)$ satisfies $f_j(\omega) \geq c_0$ on the unit sphere. Therefore,

$$P(v^{(j)} \in \mathcal{W}|Y, X) = \int_{\mathcal{W}} f_j(\omega)d\tau(\omega) \geq c_0\tau(\mathcal{W}) > 0.$$

By conditional independence across $j \in \mathcal{J}$,

$$P(\forall j \in \mathcal{J} : v^{(j)} \notin \mathcal{W} | Y, X) \leq (1 - c_0 \tau(\mathcal{W}))^{p_1} \to 0,$$

and hence

$$P(\exists j \in \mathcal{J} : v^{(j)} \in \mathcal{W} | Y, X) \to 1$$

as $p_1 \to \infty$. Finally, if $v^{(j)} \in \mathcal{W}$, then the radial section $\{r > 0 : rv^{(j)} \in \mathcal{V}\}$ has positive Lebesgue measure, and hence there exists $r > 0$ such that $rv^{(j)} \in \mathcal{V}$. Since

$$V^{(j)} = \|V^{(j)}\| v^{(j)} = \frac{\|V^{(j)}\|}{r}(rv^{(j)}),$$

the variable $V^{(j)}$ is a positive scalar multiple of a vector in $\mathcal{V}$. Replacing a regressor by a positive multiple does not change the column space spanned by $(X, V)$, and therefore leaves the coefficients on $X$ and their signs unchanged. It follows that $V^{(j)}$ also satisfies the sign relation in (3.3). This proves the theorem. ∎

**Proof of Remark 3.2.** Recall that $\hat{\alpha}_V = (V' M_X V)^{-1} V' M_X Y$ and under the construction $V = XA + bY + \varepsilon \vec{u}$, we have $M_X V = b M_X Y + \varepsilon M_X \vec{u}$. Substituting yields

$$\hat{\alpha}_V = \frac{b Y' M_X Y + \varepsilon Y' M_X \vec{u}}{b^2 Y' M_X Y + 2b\varepsilon Y' M_X \vec{u} + \varepsilon^2 \vec{u}' M_X \vec{u}},$$

where the denominator is strictly positive because $V' M_X V = \|M_X V\|^2 > 0$ as $V$ does not belong to the column space of $X$. Let $K_1 = Y' M_X Y = \|M_X Y\|^2$ and $K_2 = Y' M_X \vec{u}$. Then

$$\hat{\alpha}_V = \frac{b K_1 + \varepsilon K_2}{V' M_X V}.$$

Since $M_X$ is an orthogonal projection and $\|\vec{u}\| = 1$, we have $|K_2| = |Y' M_X \vec{u}| \leq \|M_X Y\|$. Moreover, $\varepsilon \leq \eta \|M_X Y\| \kappa_{\min}(A)$ along with $\kappa_{\min}(A) \leq \kappa_{\max}(A) \leq b$ lead to $\varepsilon \leq \eta b \|M_X Y\|$. It follows that

$$b K_1 + \varepsilon K_2 \geq b \|M_X Y\|^2 - \varepsilon \|M_X Y\| \geq (1 - \eta) b \|M_X Y\|^2.$$

Since $\eta \in (0, 1/4)$, the right-hand side is strictly positive. Hence $\hat{\alpha}_V > 0$, which under the normalization $b > 0$ is equivalent to

$$\operatorname{sgn} \hat{\alpha}_V = \operatorname{sgn} \frac{1}{b}.$$

This proves the claim. ∎

**Proof of Theorem 3.2.** Let $Z = [X, V]$ and $M_{XV} = I - Z(Z'Z)^{-1} Z'$. By the partitioned inverse identity,

$$M_{XV} = M_X - \frac{M_X V V' M_X}{V' M_X V},$$

so that

$$\hat{\sigma}^2 = \frac{1}{n - q - 1} \left( Y' M_X Y - \frac{(Y' M_X V)^2}{V' M_X V} \right).$$

Under the construction $V = XA + bY + \varepsilon \vec{u}$, noting that $M_X X = 0$, we have $M_X V = b M_X Y + \varepsilon M_X \vec{u}$. Define $K_1 = Y' M_X Y$, $K_2 = Y' M_X \vec{u}$ and $K_3 = \vec{u}' M_X \vec{u}$. Then

$$\hat{\sigma}^2 = \frac{1}{n - q - 1} \frac{\varepsilon^2 (K_1 K_3 - K_2^2)}{V' M_X V}.$$

By the Cauchy–Schwarz inequality, $K_1 K_3 - K_2^2 \geq 0$, with strict inequality for almost every $\vec{u} \in \mathbb{S}^{n-1}$. Moreover,

$$V' M_X V \geq b^2 \|M_X Y\|^2 (1 - 2\eta) > 0$$

where the inequality follows from $|K_2| \leq \|M_X Y\|$, $K_3 \leq 1$ and the constraint on $\varepsilon$. It follows that

$$\hat{\sigma}^2 \leq C_\sigma \varepsilon^2$$

for some constant $C_\sigma > 0$.

Next, by the block inverse formula,

$$(Z'Z)^{-1} = \begin{bmatrix} (X'X)^{-1} + \dfrac{h(\varepsilon)h(\varepsilon)'}{V'M_X V} & * \\ * & * \end{bmatrix}, \qquad h(\varepsilon) = (X'X)^{-1} X' V.$$

Since $h(\varepsilon) = A + b\hat{\beta} + \varepsilon (X'X)^{-1} X' \vec{u}$, it is uniformly bounded for all $\vec{u} \in \mathbb{S}^{n-1}$ and $0 < \varepsilon < \varepsilon_0$, and hence

$$[(Z'Z)^{-1}]_{jj} \leq C_h, \qquad se(\hat{\alpha}_{X,j}) \leq C' \varepsilon.$$

By the proof of Theorem 2.1, for all admissible $V \in \mathcal{V}$,

$$|\hat{\alpha}_{X,j}| \geq \frac{1 - 3\eta - 2\eta^2}{1 + 2\eta + \eta^2} \frac{|A_j|}{b} = C_{\alpha,j} > 0.$$

Since $q$ is fixed, it follows that

$$\min_{j \in [q]} |T_j| = \min_{j \in [q]} \frac{|\hat{\alpha}_{X,j}|}{se(\hat{\alpha}_{X,j})} \geq \frac{C_*}{\varepsilon}$$

for some $C_* > 0$.

For any $c > 0$ and $\delta \in (0,1)$, define $\varepsilon_1(c,\delta) = \frac{C_*}{c \cdot t_{n-q-1,\delta/2}}$ and let $\varepsilon_1^* = \min\{\varepsilon_0, \varepsilon_1(c,\delta)\}$. Consider

$$\mathcal{V}_{\delta,c} = \{V \in \mathcal{V} : 0 < \varepsilon \leq \varepsilon_1^*\}.$$

Then $\mathcal{V}_{\delta,c}$ has positive Lebesgue measure, since it is a translation of the punctured ball $\{w \in \mathbb{R}^n : 0 < \|w\| \leq \varepsilon_1^*\}$, and every $V \in \mathcal{V}_{\delta,c}$ satisfies

$$\min_{j \in [q]} |T_j| \geq c \cdot t_{n-q-1,\delta/2}.$$

The final probabilistic statement follows by applying the same directional-measure and positive-rescaling argument as in Theorem 3.1 to the positive-measure set $\mathcal{V}_{\delta,c}$. ∎

***Proof of Corollary 3.2.1.*** From the block inverse formula, we have $\widehat{Var}(\hat{\alpha}_V) = \hat{\sigma}^2 (V' M_X V)^{-1}$. By the proof of Theorem 3.2,

$$\hat{\sigma}^2 \leq C_\sigma \varepsilon^2, \qquad V' M_X V \geq b^2 \|M_X Y\|^2 (1 - 2\eta) > 0.$$

Hence $se(\hat{\alpha}_V) = \sqrt{Var(\hat{\alpha}_V)} \leq C_V' \varepsilon$ for some constant $C_V' > 0$.

Next, by the proof of Remark 3.2,

$$\hat{\alpha}_V = \frac{bK_1 + \varepsilon K_2}{V' M_X V}, \qquad K_1 = \|M_X Y\|^2, \quad K_2 = Y' M_X \vec{u}.$$

Since $|K_2| \leq \|M_X Y\|$, the worst case occurs when $K_2 = -\|M_X Y\|$, and using $\varepsilon \leq \eta b \|M_X Y\|$, we obtain

$bK_1 + \varepsilon K_2 \geq (1-\eta)b\|M_X Y\|^2$. Moreover, $V'M_X V \leq (1 + 2\eta + \eta^2)b^2\|M_X Y\|^2$. Therefore

$$\hat{\alpha}_V \geq \frac{1-\eta}{1+2\eta+\eta^2}\frac{1}{b} = C_V > 0, \qquad \text{sgn } \hat{\alpha}_V = \text{sgn } \frac{1}{b}.$$

It follows that

$$|T_V| = \frac{|\hat{\alpha}_V|}{se(\hat{\alpha}_V)} \geq \frac{C_V}{C_V'\varepsilon} = \frac{C_2}{\varepsilon}$$

for some constant $C_2 > 0$.

For any $c > 0$ and $\delta \in (0,1)$, define

$$\varepsilon_2(c,\delta) = \frac{C_2}{c \cdot t_{n-q-1,\delta/2}}, \qquad \varepsilon_2^* = \min\{\varepsilon_0, \varepsilon_1(c,\delta), \varepsilon_2(c,\delta)\},$$

and let

$$\mathcal{V}_{\delta,c}^1 = \{V \in \mathcal{V} : 0 < \varepsilon \leq \varepsilon_2^*\}.$$

Since $\varepsilon_2^* \leq \varepsilon_1^*$, we have $\mathcal{V}_{\delta,c}^1 \subseteq \mathcal{V}_{\delta,c}$. Moreover, $\mathcal{V}_{\delta,c}^1$ has positive Lebesgue measure, since it is a translation of the punctured ball $\{w \in \mathbb{R}^n : 0 < \|w\| \leq \varepsilon_2^*\}$, and every $V \in \mathcal{V}_{\delta,c}^1$ satisfies

$$|T_V| \geq c \cdot t_{n-q-1,\delta/2},$$

while the conclusions of Theorem 3.2 continue to hold.

The final probabilistic statement follows by applying the same directional-measure and positive-rescaling argument as in Theorem 3.1 to the positive-measure set $\mathcal{V}_{\delta,c}^1$. ∎

***Proof of Corollary 3.2.2.*** Let $\hat{\alpha}_{X,I}$ denote the subvector indexed by $I$, and let $\hat{\Sigma}_I = \widehat{\text{Var}}(\hat{\alpha}_{X,I})$. The $F$-statistic is

$$F_I = \frac{1}{s}\hat{\alpha}'_{X,I}\hat{\Sigma}_I^{-1}\hat{\alpha}_{X,I}.$$

By the proof of Theorem 3.2, for each $j \in [q]$, there exists a constant $C_{\alpha,j} > 0$ such that $|\hat{\alpha}_{X,j}| \geq C_{\alpha,j}$. Since $q$ is fixed, $C_\alpha = \min_{j \in [q]} C_{\alpha,j} > 0$, and hence all components of $\hat{\alpha}_{X,I}$ are uniformly bounded away from zero, so that

$$\|\hat{\alpha}_{X,I}\|^2 \geq sC_\alpha^2.$$

By the proof of Theorem 3.2, the $X$-block $S_1$ of $(Z'Z)^{-1}$ is uniformly bounded in operator norm over $\mathcal{V}$. Hence, for the principal submatrix $S_{1,II}$, there exists a constant $C_h > 0$ such that

$$S_{1,II} \preceq C_h I_s,$$

where $\preceq$ denotes the Loewner order on symmetric matrices.

Moreover, by the proof of Theorem 3.2, $\hat{\sigma}^2 \leq C_\sigma \varepsilon^2$. Since $\hat{\Sigma}_I = \hat{\sigma}^2 S_{1,II}$, it follows that

$$\hat{\Sigma}_I \preceq C_2\varepsilon^2 I_s, \qquad \hat{\Sigma}_I^{-1} \succeq \frac{1}{C_2\varepsilon^2}I_s,$$

by standard eigenvalue monotonicity, where $C_2 = C_\sigma C_h$.

Therefore

$$F_I \geq \frac{1}{s} \cdot \frac{1}{C_2\varepsilon^2}\|\hat{\alpha}_{X,I}\|^2 \geq \frac{C_\alpha^2}{C_2} \cdot \frac{1}{\varepsilon^2}.$$

For any $c > 0$ and $\delta \in (0, 1)$, define

$$\varepsilon_3(c, \delta) = \sqrt{\frac{C_\alpha^2}{C_2 \cdot c \cdot F_{\delta;s,n-q-1}}}, \qquad \varepsilon_3^* = \min\{\varepsilon_0, \varepsilon_1(c, \delta), \varepsilon_2(c, \delta), \varepsilon_3(c, \delta)\},$$

and let

$$\mathcal{V}_{\delta,I,c} = \{V \in \mathcal{V} : 0 < \varepsilon \leq \varepsilon_3^*\}.$$

Since $\varepsilon_3^* \leq \varepsilon_2^*$, we have $\mathcal{V}_{\delta,I,c} \subseteq \mathcal{V}_{\delta,c}^1$. Moreover, $\mathcal{V}_{\delta,I,c}$ has positive Lebesgue measure, since it is a translation of the punctured ball $\{w \in \mathbb{R}^n : 0 < \|w\| \leq \varepsilon_3^*\}$, and every $V \in \mathcal{V}_{\delta,I,c}$ satisfies

$$F_I \geq c \cdot F_{\delta;s,n-q-1}.$$

The final probabilistic statement follows by applying the same directional-measure and positive-rescaling argument as in Theorem 3.1 to the positive-measure set $\mathcal{V}_{\delta,I,c}$. ∎

# B   Supplementary geometric characterization

For any observational study, the best linear projection (BLP) offers an interpretable measure of association, regardless of the true underlying functional form. We begin by illustrating why a sign reversal may occur by examining the BLP within a vector space framework.

Let $\vec{y}, \vec{x} \in \mathbb{R}^n$. The orthogonal decomposition of $\vec{y}$ onto $\vec{x}$ is

$$\vec{y} = \vec{x}\hat{\beta}_x + \vec{e}_x,$$

where $\hat{\beta}_x = \arg\min_\beta \|\vec{y} - \vec{x}\beta\|^2$ is the BLP coefficient and $\vec{e}_x$ is the residual vector. Suppose an additional vector $\vec{v} \in \mathbb{R}^n$ is observed, leading to the expanded decomposition

$$\vec{y} = \vec{x}\hat{\alpha}_x + \vec{v}\hat{\alpha}_v + \vec{e}_{xv},$$

where $(\hat{\alpha}_x, \hat{\alpha}_v)' = \arg\min_{\alpha_x, \alpha_v} \|\vec{y} - \vec{x}\alpha_x - \vec{v}\alpha_v\|^2$ and $\vec{e}_{xv}$ is the residual vector.

The following lemma characterizes the exact geometric condition under which $\hat{\beta}_x$ and $\hat{\alpha}_x$ have opposite signs when vectors $\vec{y}, \vec{x}$ and $\vec{v}$ lie in the same plane.

**Lemma B.1.** *Let $\vec{y}, \vec{x} \in \mathbb{R}^n$ and suppose $\vec{v}$ lies in the plane $\mathrm{span}\{\vec{y}, \vec{x}\}$. Then $\mathrm{sgn}\,\hat{\beta}_x = -\mathrm{sgn}\,\hat{\alpha}_x$ if and only if*

$$\cos\theta(\cos\theta - \cos\delta\cos\phi) < 0, \tag{B.1}$$

*where $\theta$ is the angle between $\vec{y}$ and $\vec{x}$, $\delta$ is the angle between $\vec{x}$ and $\vec{v}$, and $\phi$ is the angle between $\vec{y}$ and $\vec{v}$.*

The condition in Eq. (B.1) admits a clear geometric interpretation and can be readily verified across the different configurations shown in Figure 2.

To characterize all possible orientations of $\vec{v}$ when it does not lie in the plane $\mathrm{span}\{\vec{y}, \vec{x}\}$, consider an arbitrary plane $\mathcal{P}$ containing $\vec{x}$ but distinct from $\mathrm{span}\{\vec{y}, \vec{x}\}$. For any $\vec{v} \in \mathcal{P}$ that is not collinear with $\vec{x}$, rotating $\mathcal{P}$ around $\vec{x}$ traces out all admissible orientations of $\vec{v}$. The following proposition extends Lemma B.1 to this general case, providing a necessary and sufficient condition for sign reversal when $\vec{v}$ lies outside the plane $\mathrm{span}\{\vec{y}, \vec{x}\}$.

**Proposition B.1.** *Let $\vec{y}$ and $\vec{x}$ span the plane $\mathrm{span}\{\vec{y}, \vec{x}\}$. Consider any plane $\mathcal{P}$ containing $\vec{x}$ but distinct from $\mathrm{span}\{\vec{y}, \vec{x}\}$, and let $\vec{v} \in \mathcal{P}$ be noncollinear with $\vec{x}$. Define*

- *$\theta$: the angle between $\vec{y}$ and $\vec{x}$,*

- *$\gamma \in (0, \pi/2)$: the acute angle between $\vec{y}$ and $\mathcal{P}$,*

- $\alpha \in (0, \pi)$: *the dihedral angle between* $\mathcal{P}$ *and* $\mathrm{span}\{\vec{y}, \vec{x}\}$, *and*

- $\eta$: *the angle between* $\vec{x}$ *and the projection of* $\vec{y}$ *onto* $\mathcal{P}$.

*Then*

$$\sin \gamma = \sin \theta \sin \alpha, \tag{B.2}$$

$$\cos \gamma = \cos \theta \cos \eta + \sin \theta \sin \eta \cos \alpha, \tag{B.3}$$

*and* $\eta$ *is uniquely identified. Furthermore, let* $\delta$ *be the angle between* $\vec{x}$ *and* $\vec{v}$, *and let* $\phi$ *be the angle between* $\vec{v}$ *and the projection of* $\vec{y}$ *onto* $\mathcal{P}$. *A sign reversal* $\mathrm{sgn}\,\hat{\beta}_x = -\,\mathrm{sgn}\,\hat{\alpha}_x$ *occurs if and only if*

$$\cos \eta (\cos \eta - \cos \delta \cos \phi) < 0. \tag{B.4}$$

The identifiability of the angle $\eta$ is central to Proposition B.1. For any fixed $\vec{y}$ and plane $\mathcal{P}$, $\eta$ is uniquely determined. Once $\eta$ is fixed, the projection of $\vec{y}$ onto $\mathcal{P}$, together with $\vec{x}$ and $\vec{v}$, lies within the same plane, reducing the problem to the planar case of Lemma B.1. As the dihedral angle $\alpha \to 0$, $\eta \to \theta$, recovering Lemma B.1. Conversely, as $\alpha \to \pi/2$, $\eta \to 0$, and no $\vec{v} \in \mathcal{P}$ can induce a sign reversal to $\hat{\beta}_x$.

Taken together, the geometric intuition in Figure 2, Lemma B.1 and Proposition B.1 imply that when the vectors $\vec{y}$ and $\vec{x}$ form an acute angle, any auxiliary vector whose direction lies within this acute angular sector, or equivalently within its vertical angle, induces a sign change in the coefficient on $\vec{x}$. When $\vec{y}$ and $\vec{x}$ form an obtuse angle, any auxiliary vector whose direction lies within the supplementary angular sector of this obtuse angle, or equivalently within its vertical angle, likewise leads to a sign change in the coefficient on $\vec{x}$.

## B.1 Proof of Lemma B.1 and Proposition B.1

***Proof of Lemma B.1.*** By the definition of the angle between two nonzero vectors,

$$\cos \theta = \frac{\langle \vec{y}, \vec{x} \rangle}{\sqrt{\langle \vec{y}, \vec{y} \rangle \langle \vec{x}, \vec{x} \rangle}}, \quad \cos \delta = \frac{\langle \vec{x}, \vec{v} \rangle}{\sqrt{\langle \vec{v}, \vec{v} \rangle \langle \vec{x}, \vec{x} \rangle}}, \quad \cos \phi = \frac{\langle \vec{y}, \vec{v} \rangle}{\sqrt{\langle \vec{y}, \vec{y} \rangle \langle \vec{v}, \vec{v} \rangle}}.$$

For the projection of $\vec{y}$ onto $\vec{x}$, the projection coefficient is $\hat{\beta}_x = \frac{\langle \vec{y}, \vec{x} \rangle}{\langle \vec{x}, \vec{x} \rangle}$, so that $\cos \theta = \hat{\beta}_x \sqrt{\frac{\langle \vec{x}, \vec{x} \rangle}{\langle \vec{y}, \vec{y} \rangle}}$.

By the Frisch-Waugh-Lovell theorem, the partial effect of $\vec{x}$ after controlling for $\vec{v}$ is proportional to $\hat{\alpha}_x \propto \langle \vec{y} - \Pi_{\vec{v}}\vec{y}, \vec{x} - \Pi_{\vec{v}}\vec{x} \rangle$, where $\Pi_{\vec{v}}$ denotes the orthogonal projection onto $\mathrm{span}(\vec{v})$. Since

$$\Pi_{\vec{v}}\vec{y} = \vec{v}\sqrt{\frac{\langle \vec{y}, \vec{y} \rangle}{\langle \vec{v}, \vec{v} \rangle}} \cos \phi, \quad \Pi_{\vec{v}}\vec{x} = \vec{v}\sqrt{\frac{\langle \vec{x}, \vec{x} \rangle}{\langle \vec{v}, \vec{v} \rangle}} \cos \delta,$$

it follows that

$$\langle \vec{y} - \Pi_{\vec{v}}\vec{y}, \vec{x} - \Pi_{\vec{v}}\vec{x} \rangle = \sqrt{\langle \vec{y}, \vec{y} \rangle \langle \vec{x}, \vec{x} \rangle}(\cos \theta - \cos \phi \cos \delta).$$

Hence $\hat{\alpha}_x \propto \cos \theta - \cos \phi \cos \delta$, while $\hat{\beta}_x \propto \cos \theta$, implying

$$\mathrm{sgn}\,\hat{\beta}_x = -\,\mathrm{sgn}\,\hat{\alpha}_x \quad \text{iff} \quad \cos \theta(\cos \theta - \cos \phi \cos \delta) < 0.$$

∎

***Proof of Proposition B.1.*** The identities (B.2) and (B.3) follow directly from classical spherical trigonometry; see, for example, Palmer and Leigh (1934, p. 196).

To identify $\eta$, observe that $\alpha$ and $\theta$ are fixed by the observed vectors $\vec{y}, \vec{x}$ and the plane $\mathcal{P}$. Since $\gamma$ is the acute angle between $\vec{y}$ and $\mathcal{P}$, Eq. (B.2) uniquely determines $\gamma \in (0, \pi/2)$. Substituting $\cos^2 \eta = 1 - \sin^2 \eta$ into

Eq. (B.3) yields a quadratic equation in $\sin\eta$:

$$A\sin^2\eta - 2B\sin\eta + C = 0,$$

where

$$A = \sin^2\theta\cos^2\alpha + \cos^2\theta, \quad B = \sin\theta\cos\alpha\cos\gamma, \quad C = \cos^2\gamma - \cos^2\theta.$$

The discriminant is

$$\Delta = 4B^2 - 4AC = 4(\sin^2\theta\cos^2\alpha\cos^2\theta - \cos^2\theta\cos^2\gamma + \cos^4\theta). \tag{B.5}$$

Using the identity $\sin\gamma = \sin\theta\sin\alpha$ from (B.2) implies $\cos^2\gamma = \cos^2\theta + \sin^2\theta\cos^2\alpha$, which yields $\Delta = 0$ and hence a unique real solution,

$$\sin\eta = \frac{B}{A} = \frac{\sin\theta\cos\alpha\cos\gamma}{\sin^2\theta\cos^2\alpha + \cos^2\theta} \leq \left(\frac{\sin^2\theta\cos^2\alpha}{\sin^2\theta\cos^2\alpha + \cos^2\theta}\right)^{1/2} < 1$$

where the inequality follows from the Cauchy-Schwarz inequality and the identity (B.2). Since all terms are strictly positive, $\sin\eta$ is uniquely identified in $(0,1)$.

A projection argument further gives $\cos\eta = \frac{\cos\theta}{\cos\gamma}$, as $\vec{x} \in \mathcal{P}$ is orthogonal to the component of $\vec{y}$ perpendicular to $\mathcal{P}$. Given $\gamma > 0$, both $\sin\eta$ and $\cos\eta$ are uniquely determined, thereby identifying $\eta$.

Finally, within $\mathcal{P}$, the vectors $\vec{x}$, $\vec{v}$ and the projection of $\vec{y}$ onto $\mathcal{P}$ are coplanar, so Lemma B.1 applies in $\mathcal{P}$ with $\theta$ replaced by $\eta$, yielding $\cos\eta(\cos\eta - \cos\delta\cos\phi) < 0$. ∎

## C  Connections to related phenomena

This section relates SHAVE to several well-documented empirical and behavioral phenomena.

**Spurious correlation.** SHAVE can be interpreted as a manifestation of spurious correlation, where statistical association arises without a causal link. Such correlations may result from latent confounders or sampling variation and have been documented in both low- (Aldrich, 1995) and high-dimensional (Fan and Lv, 2008; Fan et al., 2014) settings. In our framework, although the auxiliary variable $V$ has no causal effect on $Y$, it may exhibit nonzero sample correlations with both $Y$ and $X$. When $V$ falls within the sign change set, these correlations can induce coefficient reversals in finite samples. This mechanism is consistent with classical examples, including spurious significance in time series driven by shared shocks or trends. In all cases, the observed relationships are induced by sample geometry rather than structural dependence.

**The $p$-hacking phenomenon.** SHAVE provides a formal mechanism underlying $p$-hacking, defined as the selective reporting of statistically significant results. Theorem 3.2 shows that, appropriately chosen auxiliary variables can generate arbitrarily large $t$-statistics, even in the absence of a true association between $X$ and $Y$. Selective reporting of such specifications contributes to publication bias across disciplines (Simmons et al., 2011; Simonsohn et al., 2014; Head et al., 2015; Brodeur et al., 2020; Lindsay, 2015; Elliott et al., 2022). This perspective highlights how apparently robust findings may arise from specification search rather than underlying signal.

**Sponsorship bias.** SHAVE also sheds light on sponsorship bias, where empirical conclusions systematically favor particular interests (Lesser et al., 2007). The existence of a positive-measure sign change set implies that favorable results can, in principle, be obtained through variable exploration, even in large samples. A prominent example is the Harvard admissions case, where opposing experts reach contradictory conclusions about racial effects (Arcidiacono, 2018; Card, 2018), illustrating how model specification can align with differing incentives. Similar patterns have been documented in industry-funded research, including studies on sugar (Kearns et al., 2016), tobacco (Barnes and Bero, 1998), nutrition (Lesser et al., 2007) and biomedical outcomes (Bekelman et al., 2003). These examples suggest that the flexibility in model specification can contribute to systematic bias under conflicting interests.

**Research misconduct.** In extreme cases, SHAVE corresponds to research misconduct, including falsification

or fabrication of empirical results. Recent studies estimate that roughly 3–8% of researchers have engaged in such practices (Xie et al., 2021; Gopalakrishna et al., 2022). In political science, Lenz and Sahn (2021) show that many published findings hinge on undisclosed specification choices. The examples in Section 4.3 further illustrate how easily sign hacking can generate convincing yet misleading results in large datasets.

Artificially significant findings can distort the scientific record and undermine credibility. High-profile cases, including Dr. Piero Anversa's falsified stem-cell research (Kolata, 2018), Dr. Erin Potts-Kant's misconduct at Duke University (Jonathan, 2019), and the STAP cell scandal (Cryanoski, 2014), illustrate the broader consequences of empirical manipulation. These examples underscore the importance of transparency in empirical analysis for maintaining research integrity.

# D   Additional empirical evidence

We provide further evidence on the feasibility and ease of implementing SHAVE in practice. Following the same setup as in the main analysis, we set probe 1389163_at as $Y$ and let a single probe (1369353_at) or two probes (1369353_at and 1370429_at) from the 24 TRIM32-associated probes form $X_{\mathcal{I}_1}$. The control variables $X_{\mathcal{C}}$ are drawn from the remaining associated probes with cardinality $|\mathcal{C}| \in \{6, \ldots, 16\}$, and are randomly sampled up to 10,000 times per size, yielding 110,000 specifications in total. For each specification, we estimate the baseline regression of $Y$ on $X_{\mathcal{I}_1}$ and $X_{\mathcal{C}}$ and search over the remaining 19,007 probes for an auxiliary variable $V$ that induces a sign reversal in $X_{\mathcal{I}_1}$ while maintaining statistical significance at the 10% level. If simultaneous sign reversal is infeasible, we require that at least one coefficient reverses sign while both post-inclusion estimates remain significant.

Figure 3 reports, for each control set size, both the number of candidate variables $V$ and the corresponding number of control variable combinations. Across all configurations, multiple variables $V$ consistently induce sign reversals, with a mild decline as the control set size increases. Moreover, each such variable typically operates across multiple specifications with the same control size.

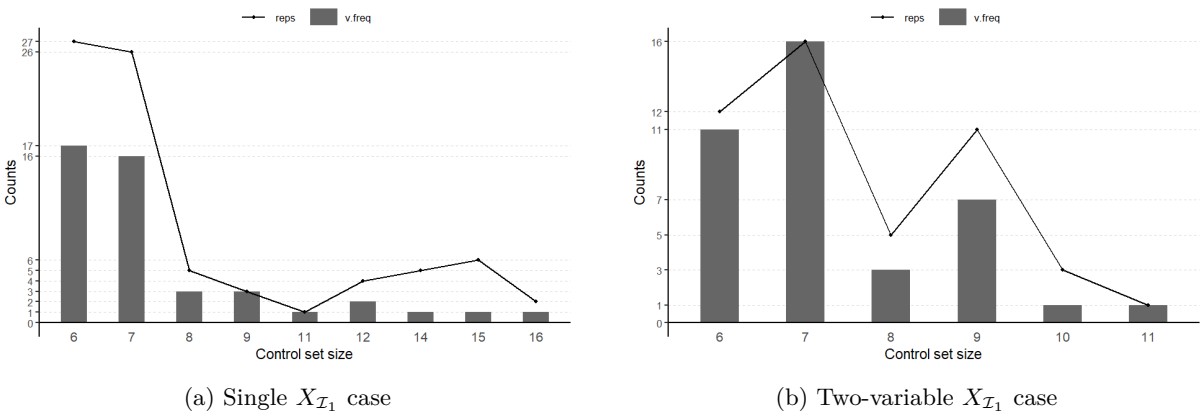

(a) Single $X_{\mathcal{I}_1}$ case      (b) Two-variable $X_{\mathcal{I}_1}$ case

Figure 3: Number of selected $V$s and corresponding model specifications across control set sizes

Table 5 presents representative regression results illustrating that a given variable $V$ can induce sign changes across different control sets. When $X_{\mathcal{I}_1}$ contains only probe 1369353_at, baseline estimates are consistently negative across specifications, whereas including $V$ yields significantly positive coefficients at the 10% level. When $X_{\mathcal{I}_1}$ includes two probes, both baseline estimates are negative and insignificant, while the inclusion of $V$ produces a significantly negative coefficient for 1369353_at and a significantly positive coefficient for 1370429_at. In addition, models including $V$ consistently achieve lower BIC values and $F$-tests reject the null that the coefficient on $V$ is zero, providing conventional statistical support for its inclusion.

Taken together, these findings provide empirical evidence for the presence of SHAVE in practice. Because multiple specifications admit auxiliary variables that induce sign reversals while remaining statistically well-supported, selective reporting of such specifications can produce internally consistent yet potentially misleading conclusions. For instance, a specification including $V$ may be presented as the primary model, with alternative specifications serving as robustness checks, while maintaining stable signs and significance levels across reported

Table 5: Sign change phenomenon for $X_{\mathcal{I}_1}$

| Cat. | Var. | M1 | M2 | M3 | M4 | M5 | M6 |
|---|---|---|---|---|---|---|---|
| **Panel A. Single $X_{\mathcal{I}_1}$ Case** | | | | | | | |
| $X_{\mathcal{I}_1}$ | 1369353_at | -0.027 | 0.236* | -0.029 | 0.222* | -0.002 | 0.223* |
| | | (0.123) | (0.133) | (0.120) | (0.130) | (0.118) | (0.126) |
| $V$ | 1372894_at | | 0.301*** | | 0.286*** | | 0.276*** |
| | | | (0.075) | | (0.074) | | (0.073) |
| $|\mathcal{C}|$ | | 12 | 12 | 14 | 14 | 15 | 15 |
| BIC | | 257.85 | 245.85 | 258 | 246.59 | 255.08 | 244.13 |
| **Panel B. Two-variable $X_{\mathcal{I}_1}$ Case** | | | | | | | |
| $X_{\mathcal{I}_1}$ | 1369353_at | -0.151 | $-0.183^{**}$ | -0.142 | $-0.176^{*}$ | -0.139 | $-0.158^{*}$ |
| | | (0.094) | (0.088) | (0.106) | (0.098) | (0.093) | (0.087) |
| | 1370429_at | -0.037 | 0.194* | -0.026 | 0.204* | -0.003 | 0.194* |
| | | (0.110) | (0.117) | (0.118) | (0.119) | (0.107) | (0.110) |
| $V$ | 1375833_at | | $-0.341^{***}$ | | $-0.359^{***}$ | | $-0.316^{***}$ |
| | | | (0.081) | | (0.077) | | (0.076) |
| $|\mathcal{C}|$ | | 7 | 7 | 8 | 8 | 9 | 9 |
| BIC | | 240.81 | 227.9 | 246.78 | 229.82 | 237.54 | 224.34 |

Notes: Standard errors are reported in parentheses. $^{***}p < 0.01$, $^{**}p < 0.05$, $^{*}p < 0.1$. $|\mathcal{C}|$ denotes the number of controls.

Table 6: Choice and selection frequencies of $V$ for specific $X_{\mathcal{I}_1}$

| **Panel A. Single-variable specification** | | | | | |
|---|---|---|---|---|---|
| $X_{\mathcal{I}_1}$ | 1370429_at | 1393979_at | 1381787_at | 1383673_at | 1393382_at |
| $V$ (freq.) | 1008 | 282 | 1171 | 1761 | 774 |
| **Panel B. Two-variable specification** | | | | | |
| $X_{\mathcal{I}_1}$ | 1370429_at | 1370429_at | 1370429_at | 1370429_at | 1393979_at |
| | 1393979_at | 1381787_at | 1383673_at | 1393382_at | 1381787_at |
| $V$ (freq.) | 52 | 43 | 202 | 244 | 26 |
| $X_{\mathcal{I}_1}$ | 1393979_at | 1393979_at | 1381787_at | 1381787_at | 1383673_at |
| | 1383673_at | 1393382_at | 1383673_at | 1393382_at | 1393382_at |
| $V$ (freq.) | 61 | 76 | 142 | 150 | 167 |

Notes: Row $X_{\mathcal{I}_1}$ lists the variables included in the specification. "$V$ (freq.)" reports the number of candidate $V$s that induce a statistically significant sign reversal when included from 19,007 probes.

models. Since the true data-generating process is unobserved, this intrinsic model uncertainty creates scope for strategic manipulation.

We conduct another analysis to illustrate the potential for SHAVE. We again take probe 1389163_at as the response variable $Y$. More generally, we select $m = 1$ or $m = 2$ probes from the 24 TRIM32-associated probes to form $X_{\mathcal{I}_1}$, representing variables with desired signs, and use the remaining probes (23 or 22) as controls $X_{\mathcal{I}_1^c}$. We estimate the baseline regression of $Y$ on $X_{\mathcal{I}_1}$ and $X_{\mathcal{I}_1^c}$ and then search among the remaining 19,007 probes for variables $V$, whose inclusion reverses the signs of the coefficients in $X_{\mathcal{I}_1}$.

A probe $V$ is retained if it satisfies two criteria: (i) in the regression of $Y$ on $X_{\mathcal{I}_1}$ and $V$, the coefficients of $X_{\mathcal{I}_1}$ reverse sign relative to the baseline and are significant at the 5% level; and (ii) in the regression including $X_{\mathcal{I}_1^c}$, the reversed signs persist.

Table 6 reports the probes $V$ that satisfying these criteria. When $X_{\mathcal{I}_1}$ contains a single variable, multiple probes are hackable, each associated with a large number of candidate auxiliary variables. When $X_{\mathcal{I}_1}$ contains two variables, several probe pairs admit multiple auxiliary variables capable of inducing simultaneous sign reversals.

Table 7 presents representative regression results. Model M1 reports the baseline regression. M2 augments the model with $V$, M3 includes all controls, M4 retains a subset of controls preserving the sign reversal, and M5 reports the marginal regression. In each panel, $V$ is selected to minimize BIC among all admissible candidates.

Table 7: Sign change phenomenon for $X_{\mathcal{I}_1}$

| Cat. | Var. | M1 | M2 | M3 | M4 | M5 |
|------|------|-----|-----|-----|-----|-----|
| **Panel A. Single $X_{\mathcal{I}_1}$ Specification** | | | | | | |
| $X_{\mathcal{I}_1}$ | 1370429_at | 0.022 | $-0.641^{***}$ | $-0.006$ | $-0.166^{**}$ | $-0.687^{***}$ |
| | | (0.125) | (0.074) | (0.118) | (0.083) | (0.067) |
| $V$ | 1383640_at | – | 0.102 | $-0.244^{***}$ | $-0.206^{***}$ | – |
| | | | (0.074) | (0.067) | (0.067) | |
| $|\mathcal{C}|$ | | 23 | – | 23 | 14 | – |
| $BIC$ | | 265.98 | 275.44 | 255.23 | 241.28 | 272.55 |
| **Panel B. Two-variable $X_{\mathcal{I}_1}$ Specification** | | | | | | |
| $X_{\mathcal{I}_1}$ | 1370429_at | 0.022 | $-0.751^{***}$ | $-0.072$ | $-0.244^{**}$ | $-0.562^{***}$ |
| | | (0.125) | (0.118) | (0.123) | (0.116) | (0.118) |
| | 1381787_at | 0.015 | $-0.278^{**}$ | $-0.018$ | $-0.206^{**}$ | $-0.151$ |
| | | (0.130) | (0.113) | (0.125) | (0.091) | (0.118) |
| $V$ | 1375775_at | – | $0.406^{***}$ | $0.287^{***}$ | $0.259^{***}$ | – |
| | | | (0.093) | (0.090) | (0.084) | |
| $|\mathcal{C}|$ | | 23 | – | 23 | 12 | – |
| $BIC$ | | 265.98 | 262.22 | 258.6 | 243.59 | 275.67 |
| $N$ | | 120 | 120 | 120 | 120 | 120 |

Notes: Standard errors are reported in parentheses. $^{***}p < 0.01$, $^{**}p < 0.05$, $^{*}p < 0.1$. $|\mathcal{C}|$ denotes the cardinality of the subset of $X_{\mathcal{I}_1^c}$ used as control variables. M1 includes all 24 probes from Huang et al. (2008). M2 adds $V$ to $X_{\mathcal{I}_1}$ without additional controls. M3 includes all 24 probes and $V$, selected by the smallest BIC value. M4 includes $V$ and a subset of $X_{\mathcal{I}_1^c}$ yielding a significant sign reversal in $X_{\mathcal{I}_1}$. M5 includes $X_{\mathcal{I}_1}$ only.

Across both the single-variable and two-variable settings, inclusion of $V$ reverses the signs of the coefficients in $X_{\mathcal{I}_1}$ and renders them statistically significant. The induced sign reversal persists after reintroducing control variables, indicating that the effect is not driven by a particular specification. For comparison, marginal regressions may yield similar signs but omit relevant covariates, and therefore do not provide a reliable benchmark.

Standard model diagnostics further favor the manipulated specifications. Models including $V$ typically achieve lower BIC values than the baseline and marginal specifications, and $F$-tests reject the null that the coefficient on $V$ is zero. These patterns are consistent across both single- and two-variable configurations, suggesting that the inclusion of $V$ is statistically well-supported under conventional criteria.

# E  Lineup detection: additional simulations

We implement the lineup detection strategy for the auxiliary variable $V$ identified in Section 4.3. In the unconditional version, each $V$ is combined with all remaining genes, including $X_{\mathcal{I}_1}$, $X_{\mathcal{I}_1^c}$ and other probes. In the conditional version, $X_{\mathcal{I}_1}$ is always included, and each $V$ is combined with $X_{\mathcal{I}_1^c}$ and the remaining probes. Lasso, SCAD and MCP are applied with cross-validated tuning parameters and repeated 50 times to account for selection instability. We summarize the selection frequency of each $V$ by reporting the proportion with positive frequency, $P(\pi_V > 0)$, and the maximum selection frequency.

Tables 8 and 9 report the results for the single-variable and two-variable specifications. Although many auxiliary variables $V$ can induce sign changes, most are rarely selected, as indicated by the uniformly small values of $P(\pi_V > 0)$ across all methods. A few isolated variables exhibit relatively large maximum selection frequencies. Selection frequencies tend to increase when conditioning on the inclusion of $X_{\mathcal{I}_1}$. Overall, these results suggest that lineup detection can be informative, but does not reliably identify all problematic auxiliary variables.

We next evaluate the ability of the detection procedure to reject spurious auxiliary variables. The data-generating process follows Model (4.1), with $n = 100$, $\beta_1 = 0.2$ and $\sigma^2 = 0.5$. We generate $1 \times 10^6$ variables from the standard normal distribution and identify a variable $V$ that induces $\mathrm{sgn}\,\hat{\beta}_1 = -\mathrm{sgn}\,\hat{\alpha}_1$ [4]. To assess

---

[4]Simulations were performed on an AMD Ryzen 9 7950X 16-Core Processor (4.50 GHz) using R version 4.4.1, one variable $V$ is identified after 923,053 iterations using random seed 14

Table 8: Lineup detection of $V$s for single-variable specification

| $X_{\mathcal{I}_1}$ | $V$ (freq.) | Lasso | | SCAD | | MCP | |
|---|---|---|---|---|---|---|---|
| | | $P(\pi_V > 0)$ | $\max(\pi_V)$ | $P(\pi_V > 0)$ | $\max(\pi_V)$ | $P(\pi_V > 0)$ | $\max(\pi_V)$ |
| Panel A. Unconditional version | | | | | | | |
| 1370429_at | 1008 | 0.026 | 1.000 | 0.006 | 1.000 | 0.009 | 1.000 |
| 1381787_at | 1171 | 0.032 | 1.000 | 0.010 | 1.000 | 0.009 | 0.980 |
| 1383673_at | 1761 | 0.020 | 1.000 | 0.005 | 1.000 | 0.005 | 0.960 |
| 1393382_at | 774 | 0.025 | 1.000 | 0.003 | 1.000 | 0.005 | 0.440 |
| 1393979_at | 282 | 0.011 | 1.000 | 0.004 | 1.000 | 0.000 | 0.000 |
| Panel B. Conditional version | | | | | | | |
| 1370429_at | 1008 | 0.030 | 1.000 | 0.018 | 1.000 | 0.007 | 1.000 |
| 1381787_at | 1171 | 0.053 | 1.000 | 0.019 | 1.000 | 0.008 | 0.900 |
| 1383673_at | 1761 | 0.035 | 1.000 | 0.021 | 1.000 | 0.010 | 1.000 |
| 1393382_at | 774 | 0.032 | 1.000 | 0.009 | 0.920 | 0.005 | 1.000 |
| 1393979_at | 282 | 0.021 | 1.000 | 0.018 | 0.800 | 0.007 | 0.500 |

Notes: For each probe $X_{\mathcal{I}_1}$, the table summarizes the selection frequencies of candidate variables $V$ that induce sign reversal. $V$ (freq.) denotes the number of such variables. For each method, $P(\pi_V > 0)$ is the proportion of $V$ with positive selection frequency across 50 repetitions, and $\max(\pi_V)$ is the maximum selection frequency among them. Panel A reports the unconditional version, while Panel B reports the conditional version.

robustness, we augment the model with additional variables $W_j, j = 1, \ldots, 499$[5], generated independently from the same distribution and not inducing sign reversal. We then construct perturbed variables $\tilde{W}$ under three settings:

**Case 1** $\tilde{W}_j = \rho \cdot W_j + \sqrt{1 - \rho^2} \cdot Z_j, j = 1, \ldots, 499$, where $\rho \in \{0.5, 0.7, 0.9\}$ and $Z_j \sim N(0, 1)$ independently. This case introduces a correlation between $\tilde{W}_j$ and the original variable $W_j$.

**Case 2** $\tilde{W}_j \sim N(0, 1)$ independently for all $j$. In this case, $\tilde{W}$ is completely independent of the original $W$.

**Case 3** $\tilde{W} \sim N(\mathbf{0}, \Sigma)$ with $\mathbf{0}$ a $499 \times 1$ zero vector and $\Sigma_{i,j} = \rho^{|i-j|}$, for $i, j = 1, \ldots, 499$ with $\rho \in \{0.5, 0.7, 0.9\}$. This structure induces dependence among the elements of $\tilde{W}$.

Table 10 presents the selection frequency of $V$ when combined with $\tilde{W}$. Across all settings, the selection frequency of $V$ remains low, indicating that the detection procedure effectively rejects spurious auxiliary variables. In Case 1, selection frequency increases moderately with $\rho$, reflecting the effect of induced correlation, but remains limited overall.

We then examine whether the detection procedure retains genuinely relevant auxiliary variables. We consider the data-generating process:

$$Y = X_1 \alpha_1 + V \alpha_2 + \epsilon,$$

where $\alpha_1 = \alpha_2 = 0.9$, and the variance of $\epsilon$ is chosen to achieve $R^2 = 0.9$. Both $X_1$ and $V$ are independently drawn from $N(0, 1)$, with $n = 100$. We generate additional variables $W_N = (W'_{N,1}, \ldots, W_{N,499})$ under three settings:

**Case 1** $W_{N,j} = \rho \cdot V + \sqrt{1 - \rho^2} \cdot Z_j$ for $j = 1$, where $Z_j$ independently follows $N(0, 1)$, $\rho \in \{0.5, 0.7, 0.9\}$, and $\rho = 0$ for $j > 1$.

**Case 2** The setting is the same as Case 1, except that $W_{N,j} = \rho \cdot V + \sqrt{1 - \rho^2} \cdot Z_j$ for $j = 1, \ldots, 5$ and $\rho = 0$ otherwise.

**Case 3** $W_{N,j} \sim N(0, 1)$ independently for all $j$.

---

[5]Including all $1 \times 10^6 - 1$ variables in $W$ is infeasible due to the sample size of 100, which may introduce additional instability in variable selection via cross-validation

Table 9: Lineup detection of $V$s for two-variable specification

| $X_{\mathcal{I}_1}$ | | $V$ (freq.) | Lasso | | SCAD | | MCP | |
|---|---|---|---|---|---|---|---|---|
| | | | $P(\pi_V > 0)$ | $\max(\pi_V)$ | $P(\pi_V > 0)$ | $\max(\pi_V)$ | $P(\pi_V > 0)$ | $\max(\pi_V)$ |
| Panel A. Unconditional version | | | | | | | | |
| 1370429_at | 1381787_at | 43 | 0.093 | 0.980 | 0.000 | 0.000 | 0.000 | 0.000 |
| 1370429_at | 1383673_at | 202 | 0.030 | 0.900 | 0.000 | 0.000 | 0.005 | 0.120 |
| 1370429_at | 1393382_at | 244 | 0.037 | 0.980 | 0.000 | 0.000 | 0.004 | 0.420 |
| 1370429_at | 1393979_at | 52 | 0.019 | 0.800 | 0.000 | 0.000 | 0.000 | 0.000 |
| 1381787_at | 1383673_at | 142 | 0.070 | 0.800 | 0.000 | 0.000 | 0.007 | 0.060 |
| 1381787_at | 1393382_at | 150 | 0.027 | 1.000 | 0.007 | 1.000 | 0.007 | 0.440 |
| 1383673_at | 1393382_at | 167 | 0.030 | 0.640 | 0.006 | 1.000 | 0.006 | 0.420 |
| 1393979_at | 1381787_at | 26 | 0.038 | 0.800 | 0.000 | 0.000 | 0.000 | 0.000 |
| 1393979_at | 1383673_at | 61 | 0.049 | 1.000 | 0.016 | 1.000 | 0.000 | 0.000 |
| 1393979_at | 1393382_at | 76 | 0.000 | 0.000 | 0.000 | 0.000 | 0.000 | 0.000 |
| Panel B. Conditional version | | | | | | | | |
| 1370429_at | 1381787_at | 43 | 0.140 | 1.000 | 0.047 | 0.840 | 0.000 | 0.000 |
| 1370429_at | 1383673_at | 202 | 0.054 | 1.000 | 0.025 | 0.900 | 0.015 | 0.800 |
| 1370429_at | 1393382_at | 244 | 0.041 | 1.000 | 0.016 | 0.960 | 0.008 | 0.840 |
| 1370429_at | 1393979_at | 52 | 0.019 | 1.000 | 0.019 | 0.920 | 0.000 | 0.000 |
| 1381787_at | 1383673_at | 142 | 0.127 | 1.000 | 0.092 | 1.000 | 0.049 | 0.440 |
| 1381787_at | 1393382_at | 150 | 0.047 | 1.000 | 0.040 | 0.740 | 0.013 | 1.000 |
| 1383673_at | 1393382_at | 167 | 0.042 | 1.000 | 0.024 | 1.000 | 0.006 | 1.000 |
| 1393979_at | 1381787_at | 26 | 0.077 | 1.000 | 0.038 | 0.560 | 0.000 | 0.000 |
| 1393979_at | 1383673_at | 61 | 0.066 | 1.000 | 0.066 | 1.000 | 0.049 | 0.900 |
| 1393979_at | 1393382_at | 76 | 0.039 | 0.880 | 0.013 | 0.760 | 0.000 | 0.000 |

Notes: For each pair of probes $X_{\mathcal{I}_1}$, the table summarizes the selection frequencies of candidate variables $V$ that induce sign reversal. $V$ (freq.) denotes the number of such variables. For each method, $P(\pi_V > 0)$ is the proportion of $V$ with positive selection frequency across 50 repetitions, and $\max(\pi_V)$ is the maximum selection frequency among them. Panel A reports the unconditional version, while Panel B reports the conditional version.

Table 10: Selection frequency of an irrelevant variable $V$

| | $\rho$ | Lasso | SCAD | MCP |
|---|---|---|---|---|
| | 0.5 | 0.29 | 0.10 | 0.09 |
| Case 1 | 0.7 | 0.26 | 0.05 | 0.10 |
| | 0.9 | 0.61 | 0.28 | 0.20 |
| Case 2 | 0 | 0.13 | 0.03 | 0.02 |
| | 0.5 | 0.16 | 0.05 | 0.04 |
| Case 3 | 0.7 | 0.27 | 0.07 | 0.08 |
| | 0.9 | 0.30 | 0.14 | 0.14 |

We incorporate the new variables $W_N$ into the model and perform variable selection to examine the selection frequency of $V$. Additionally, if $V$ is not selected, we report the proportion of cases where the estimates for $\alpha_1$ exhibit opposite signs depending on the inclusion or exclusion of $V$. Table 11 reports the selection frequency of $V$. In contrast to the previous setting, $V$ is consistently selected across all cases, supporting the ability of the detection procedure to retain relevant variables when they are part of the true model.

# F   Body fat dataset

The body fat dataset contains records for 252 men, including their percentage of body fat, age, weight, height and ten additional body circumference measurements. The objective is to examine the relationship between body fat, as an indicator of health, and the other measurements. A notable feature of this dataset is the high correlation

Table 11: Selection frequency of the auxiliary variable $V$

| | $k$ | $\rho$ | Lasso | | SCAD | | MCP | |
|---|---|---|---|---|---|---|---|---|
| | | | Sel.V | Prop.OS | Sel.V | Prop.OS | Sel.V | Prop.OS |
| Case 1 | 1 | 0.5 | 1 | 0 | 1 | 0 | 1 | 0 |
| | | 0.7 | 1 | 0 | 1 | 0 | 1 | 0 |
| | | 0.9 | 1 | 0 | 1 | 0 | 1 | 0 |
| Case 2 | 5 | 0.5 | 1 | 0 | 1 | 0 | 1 | 0 |
| | | 0.7 | 1 | 0 | 1 | 0 | 1 | 0 |
| | | 0.9 | 1 | 0 | 1 | 0 | 1 | 0 |
| Case 3 | - | - | 1 | 0 | 1 | 0 | 1 | 0 |

Note: $k$ represents the number of $W_{N,j}$'s that have correlation with $V$. Sel.V is the selection frequency of variable $V$. Prop.OS denotes the proportion of cases in which the estimate of $\alpha_1$ has opposite sign depending on the inclusion or exclusion of $V$.

among the variables. The original dataset is available on the author's website[6] or can be accessed through the R package `"mfp"`.

To ensure data quality, we exclude case 39 as an outlier (Royston and Sauerbrei, 2007) and case 42 due to apparent recording issues, resulting in a final sample size of 250. All analyses are conducted using R software. CV-based Lasso results are obtained using the *cv.glmnet* function from the `"glmnet"` package. CV-based SCAD and MCP results are obtained using the *cv.ncvreg* function from the `"ncvreg"` package. For each method, we perform 100 iterations, with each iteration yielding the final result using the CV-determined tuning parameter. The final report is compiled by combining the results from these 100 iterations.

The dependent variable in the analysis is SIRI, while the 13 independent variables are age, weight, height, neck, chest, abdomen, hip, thigh, knee, ankle, biceps, forearm and wrist. Definitions of the variables are provided in Table 12 and summary statistics for these variables are presented in Table 13.

Table 12: Definition of variables used in body fat dataset

| Var. | Definition |
|---|---|
| SIRI | Percent body fat using Brozek's equation:495/Density - 450 |
| age | Age (years) |
| weight | Weight (lbs) |
| height | Height (inches) |
| neck | Neck circumferences (cm) |
| chest | Chest circumference (cm) |
| abdomen | Abdomen circumference (cm) |
| hip | Hip circumference (cm) |
| thigh | Thigh circumference (cm) |
| knee | Knee circumference (cm) |
| ankle | Ankle circumference (cm) |
| biceps | Biceps circumference (cm) |
| forearm | Forearm circumference (cm) |
| wrist | Wrist circumference (cm) |

---

[6]http://jse.amstat.org/v4n1/datasets.johnson.html

Table 13: Summary statistics of variables used in body fat. Q25 and Q75 denote the 25% and 75% quantile.

|        | SIRI | age  | weight | height | neck | chest | abdomen | hip   | thigh | knee | ankle | biceps | forearm | wrist |
|--------|------|------|--------|--------|------|-------|---------|-------|-------|------|-------|--------|---------|-------|
| min    | 0.0  | 22.0 | 118.5  | 64.0   | 31.1 | 79.3  | 69.4    | 85.0  | 47.2  | 33.0 | 19.1  | 24.8   | 21.0    | 15.8  |
| Q25    | 12.4 | 35.3 | 158.5  | 68.3   | 36.4 | 94.3  | 84.5    | 95.5  | 56.0  | 36.9 | 22.0  | 30.2   | 27.3    | 17.6  |
| median | 19.2 | 43.0 | 176.1  | 70.0   | 38.0 | 99.6  | 90.9    | 99.3  | 59.0  | 38.5 | 22.8  | 32.0   | 28.7    | 18.3  |
| mean   | 19.0 | 44.9 | 178.1  | 70.3   | 37.9 | 100.7 | 92.3    | 99.7  | 59.2  | 38.5 | 23.1  | 32.2   | 28.7    | 18.2  |
| Q75    | 25.2 | 54.0 | 196.8  | 72.3   | 39.4 | 105.3 | 99.2    | 103.2 | 62.3  | 39.9 | 24.0  | 34.3   | 30.0    | 18.8  |
| max    | 47.5 | 81.0 | 262.8  | 77.8   | 43.9 | 128.3 | 126.2   | 125.6 | 74.4  | 46.0 | 33.9  | 39.1   | 34.9    | 21.4  |

