# OpenReview forum: "Sign Hacking with Auxiliary Variable Exploration in the Age of Big Data"
_SLADS/Section_B — Under review for SLADS_Section_B_

### Review · Reviewer_pa9t · 2026-07-14

**Summary Of Contributions:**

The paper studies the deliberate manipulation of coefficient signs through the inclusion of a chosen auxiliary variable in linear regression settings, termed SHAVE (Sign Hacking with Auxiliary Variable Exploration). The authors theoretically show that, conditional on the outcome and variables of interest, there exists a set of auxiliary-variable with positive Lebesgue measure whose inclusion can reverse the signs of existing coefficients. Additionally, with high probability, such variables can be identified from a large pool of candidates, leading to both sign reversal and inflated statistical significance. Simulation experiments and a real data application further support these theoretical findings, and strategies based on augmented or independent data are proposed to detect SHAVE.

**Audience:**

Yes

**Claims And Evidence:**

Yes

**Requested Changes:**

1. In Theorem 3.1, are $\mathcal{I}_1$ and $\mathcal{I}_2$ pre-specified, or do they vary as $p_1 \to \infty$? It would be helpful if the authors could clarify this point.
2. In the simulation studies, $10^6$ to $10^8$ auxiliary variables are generated to examine SHAVE and induce a sign change in one coefficient. Can the authors provide a theoretical result characterizing the relationship between the size of the auxiliary variable set and the probability of changing at least one coefficient sign? At a minimum, some additional discussion on this issue would be helpful.
3. In the numerical studies, all examples include only one or two covariates of interest. Can the authors provide additional examples involving more covariates of interest, particularly in high-dimensional settings where $p$ is comparable to or larger than the sample size $n$? Related to the previous comment, it would also be useful to discuss what size of auxiliary variable set would be needed in such settings.
4. It would be helpful if the authors could include simulation experiments illustrating the performance of the replication detection procedure.
5. There are some inconsistencies in notation. Although the authors state their convention for denoting vectors, this convention does not appear to be followed consistently throughout the paper. I suggest carefully reviewing and standardizing the mathematical notation, including the notation for vectors, matrices, and other mathematical objects.

**Strengths And Weaknesses:**

Strengths:
1. SHAVE is an interesting and important topic that has not been widely investigated.
2. The authors provide theoretical findings that support the study of SHAVE.
3. Numerical examples including both simulated and real-world data complement the theoretical analysis.

Weaknesses:
1. Some theoretical results need further clarification (see  requested changes below).
2. Additional simulation experiments can strengthen the work (see requested changes below).

---

### Review · Reviewer_1FYX · 2026-07-18

**Summary Of Contributions:**

This paper introduces *Sign Hacking with Auxiliary Variable Exploration* (SHAVE), a form of specification search in which an analyst selects an auxiliary regressor from a large candidate pool in order to obtain a desired sign pattern for coefficients in a baseline linear regression. The topic is timely and important because signs often determine the substantive interpretation of empirical findings, while modern data environments provide analysts with a very large number of potentially defensible transformations, controls, and extracted features.

The main theoretical contribution is a finite-sample geometric characterization of auxiliary variables that induce selective sign reversals. For a fixed outcome vector $Y$ and design matrix $X$, the authors construct auxiliary variables of the form $$V=XA+bY+\varepsilon U,$$ and show that a positive-Lebesgue-measure set of such variables can reverse the signs of any prescribed subset of the original OLS coefficients while preserving the remaining signs. This extends the usual single-coefficient sign-reversal discussion to arbitrary selective sign patterns in a multivariate regression.

The paper then connects this deterministic construction to high-dimensional search. Under a conditional richness assumption on the directions of a growing subset of candidate auxiliary variables, the probability of finding at least one variable that generates the desired sign pattern converges to one as the candidate-pool size diverges. The authors further show that the selected specification can produce arbitrarily large conventional $t$-statistics for the original regressors, a significant auxiliary-variable coefficient, and arbitrarily large conventional $F$-statistics for selected coefficient blocks.

The theory is complemented by simulation studies, an application to a rat-eye gene-expression dataset, and two proposed diagnostic strategies based on augmented-variable “lineups” and independent replication. Overall, the paper provides a novel and useful formalization of how model-search flexibility can generate apparently persuasive but selection-driven regression conclusions.

**Audience:**

Yes

**Broader Impact Concerns:**

Nothing.

**Claims And Evidence:**

Yes

**Requested Changes:**

1.  **State the nondegeneracy conditions needed for sign manipulation and conventional test statistics.**

    Condition (2.3) cannot be satisfied when $\widehat\beta_{X,j}=0$, whereas Theorem 2.1 is presently stated without excluding this case. The authors should either assume $$\widehat\beta_{X,j}\neq 0
        \quad\text{for every index whose sign is prescribed},$$ or define the target sign pattern only on the nonzero-coefficient indices. The same convention should be carried into Theorem 3.1 and the definition $S^\star\in\{-1,1\}^q$.

    For Theorem 3.2 and Corollaries 3.2.1–3.2.2, please also state $n\ge q+2$ and the rank/nondegeneracy conditions ensuring that the augmented regression is well defined and has positive residual degrees of freedom. In Appendix B, the angle-based lemmas should explicitly assume that the relevant vectors are nonzero and that the added regressor is not collinear with the baseline regressor.

2.  **Revise Assumption 1 so that it exactly supports the proof of Theorem 3.1.**

    Let $$\Omega_{j,p}=\frac{X_{A,j}}{\lVert X_{A,j}\rVert}.$$ The proof uses a conditional density $f_{j,p}(\omega\mid Y,X)$ with respect to surface measure $\tau$ and a common positive lower bound. A sufficient formulation would be $$\inf_{p}\ \inf_{j\in J_p}\ \inf_{\omega\in\mathbb S^{n-1}}
        f_{j,p}(\omega\mid Y,X)
        \ge c_0(Y,X)>0
        \quad\text{almost surely}.$$ A deterministic $c_0>0$ would be stronger but also sufficient. Without uniformity over the growing index set, the individual lower bounds could converge to zero and the product argument need not imply success probability tending to one. Please also state explicitly that the limit is taken with fixed $n$ and $q$, and clarify whether the conditional convergence holds almost surely in $(Y,X)$.

3.  **Remove the zero-residual exceptional directions from Theorem 3.2 and its corollaries.**

    The proof obtains $$\widehat\sigma^2
        =\frac{\varepsilon^2\bigl(K_1K_3-K_2^2\bigr)}
        {(n-q-1)V^\top M_XV}.$$ Equality $K_1K_3-K_2^2=0$ occurs when the relevant projected perturbation is collinear with $M_XY$. On such directions the augmented residual variance is zero, so the conventional $t$- and $F$-statistics are not properly defined. The proof correctly notes that strict inequality holds for almost every $U$, but the set $\mathcal V_{\delta,c}$ is then defined without excluding the exceptional directions and the conclusion is asserted for every $V$ in that set.

    Please define the exceptional set, for example $$\mathcal N
        =\left\{U\in\mathbb S^{n-1}:K_1K_3-K_2^2=0\right\},$$ show that $\tau(\mathcal N)=0$ under the stated rank conditions, and replace the constructed sets by their restrictions to $U\notin\mathcal N$. The resulting sets still have positive Lebesgue measure. This also supplies the positive definiteness needed when inverting $\widehat\Sigma_I$ in Corollary 3.2.2.

4. **Make the scale-invariance step explicit in the high-probability parts of Theorem 3.2 and Corollaries 3.2.1–3.2.2.**

    The directional argument identifies a vector $r\Omega_{j,p}$ in the positive-measure construction, whereas the observed candidate is a positive scalar multiple of that vector. For Theorem 3.1, invariance of the coefficients on $X$ follows from equality of the augmented column spaces. For the later results, please record explicitly that for every $a>0$, $$\widehat\alpha_X(aV)=\widehat\alpha_X(V),
        \qquad
        T_j(aV)=T_j(V),$$ while the coefficient and standard error of the auxiliary regressor both scale by $a^{-1}$, so that $$T_{aV}=T_V,$$ and the relevant block $F$-statistic is unchanged. A short lemma would make the repeated “same positive-rescaling argument” fully rigorous.

5.  **Correct the angle convention and limiting discussion in Proposition B.1.**

    Proposition B.1 defines the dihedral angle as $\alpha\in(0,\pi)$, but its proof uses $\cos\alpha>0$ when asserting positivity and when deriving the displayed solution for $\sin\eta$. The authors should either define $\alpha$ as the principal acute angle between the two planes, $\alpha\in(0,\pi/2)$ (with boundary cases treated separately), or formulate the proof using an oriented angle and carry the signs consistently. In addition, when the angle $\theta$ between $Y$ and $X$ is obtuse, $\cos\eta=\cos\theta/\cos\gamma<0$ and therefore $\eta\to\pi$, rather than $0$, as $\alpha\to\pi/2$. The no-reversal conclusion at the limiting plane may remain valid, but the geometric statement and proof should distinguish the acute and obtuse cases.

6.  **Remove, replace, or connect Lemma A.1.**

    Lemma A.1 concerns an $n$-vector with independent scalar coordinates and a density bounded below on a box. It is not invoked in the proof of Theorem 3.1, which instead assumes a density directly on the sphere. As written, the reference to this lemma in the discussion of Assumption 1 is potentially confusing. A short lemma stating the exact spherical small-set probability used in Theorem 3.1 would be more useful, or the current lemma could simply be removed.

7. **Clarify the meaning of the conventional critical values and “statistical significance.”**

    Please define the quantile notation unambiguously, for example $t_{\nu,1-\delta/2}$ and $F_{s,\nu;1-\delta}$. More importantly, Theorem 3.2 is a conditional geometric statement and does not impose the stochastic linear-model assumptions under which conventional finite-sample $t$ and $F$ reference distributions are exact. Because $V$ is selected after observing the data, the corresponding post-search $p$-values are not valid without selection adjustment. I suggest describing the results as exceeding conventional or nominal in-sample thresholds, which is fully consistent with the paper’s substantive message.

8.  **State the assumptions behind the replication calculations and slightly temper a few empirical claims.**

    The relation $$\operatorname{se}(\widetilde\beta_{X,j})
        \approx \sqrt{\frac{n_0}{n_1}}\,
        \operatorname{se}(\widehat\beta_{X,j})$$ and the sample-size formulas in Section 5.2 require comparable designs and error variances across studies, together with a normal or large-sample approximation. Please state these assumptions and explain whether uncertainty in the original estimate is intentionally conditioned upon. In the simulation discussion, “the success probability will eventually be 1” should be changed to “converges to 1.” In the gene-expression example, it would also be preferable to describe the results as demonstrating the feasibility of the mechanism rather than as evidence that actual misconduct occurred. Providing code, random seeds, and implementation details for the extremely large candidate-pool searches would further improve reproducibility.

**Strengths And Weaknesses:**

### Strengths

- **Original and societally relevant question.** The paper isolates a concrete mechanism through which researcher degrees of freedom can affect the direction, rather than only the significance, of reported effects. This gives the work a clear conceptual identity relative to the broader literatures on $p$-hacking, specification search, and model uncertainty.

- **Constructive finite-sample theory.** The positive-measure result is stronger and more informative than a statement that an isolated adversarial auxiliary variable exists. The construction also accommodates selective reversals across several coefficients rather than only a single reversal.

- **Effective bridge from geometry to large-scale search.** The directional argument in Theorem 3.1 explains why a small per-variable probability of entering a favorable region can become practically important when the candidate pool is very large.

- **The proof strategy is largely transparent.** The use of Frisch–Waugh–Lovell residualization, projection geometry, polar coordinates, and scale invariance is natural and makes the central mechanism understandable.

- **Broader inferential consequences are considered.** Showing that the same search can affect coefficient signs, conventional $t$-statistics, the auxiliary coefficient, and block $F$-statistics makes the message substantially stronger than a coefficient-instability example alone.

- **Empirical and diagnostic components add value.** The simulations and gene-expression illustration demonstrate practical relevance, while the proposed detection discussion appropriately emphasizes the importance of external information and replication.

### Weaknesses

- Several theorem statements do not yet state all nondegeneracy conditions used by their proofs. In particular, the construction in Theorem 2.1 implicitly requires nonzero baseline coefficients on every index whose sign is prescribed, and the $t$- and $F$-statistic results require positive residual degrees of freedom and nonzero residual variance.

- Assumption 1 is slightly less precise than the condition actually invoked in the proof of Theorem 3.1. The proof needs a common conditional lower bound for the directional densities across the growing candidate subset, not merely a variable-specific statement that each density is bounded away from zero.

- The proof of Theorem 3.2 acknowledges that the residual quadratic form is strictly positive only for almost every perturbation direction, but the theorem subsequently claims its conclusion for every point in the displayed set. The exceptional zero-residual directions should be removed explicitly.

- The positive-rescaling step used to transfer the high-probability directional result to the $t$- and $F$-statistic refinements is correct in spirit, but its invariance claims should be written out rather than referred to implicitly.

- The supplementary non-planar geometric result requires a correction to its angle conventions. The stated range of the dihedral angle is not consistent with positivity properties used in the proof, and one limiting statement for an obtuse baseline angle is not correct as currently written.

- The lineup procedure is best viewed as a diagnostic heuristic, and the replication formulas rely on standard-error scaling and normal or $t$ approximations that should be stated explicitly. These limitations do not undermine the main SHAVE theory, but they should be delineated more carefully.

The above issues appear local and repairable. I do not see a need for a new central theorem or a substantially expanded empirical analysis, and I would support acceptance after the proof statements and qualifications are tightened.